# Post-resolution macrophages shape long-term tissue immunity and integrity in a mouse model of pneumococcal pneumonia

Karen T. Feehan [1,7], Hannah E. Bridgewater [1,8], Jan Stenkiewicz-Witeska[1], Roel P. H. De Maeyer [2], John Ferguson[3], Matthias Mack [4], Jeremy Brown [5], Giuseppe Ercoli [5], Connar M. Mawer [5], Arne N. Akbar [1], James R. W. Glanville[1], Parinaaz Jalali[1], Olivia V. Bracken[1], Anna Nicolaou [6], Alexandra C. Kendall [6], Michelle A. Sugimoto[1] & Derek W. Gilroy [1] ✉

Resolving inflammation is thought to return the affected tissue back to homoeostasis but recent evidence supports a non-linear model of resolution involving a phase of prolonged immune activity. Here we show that within days following resolution of *Streptococcus pneumoniae*-triggered lung inflammation, there is an influx of antigen specific lymphocytes with a memory and tissue-resident phenotype as well as macrophages bearing alveolar or interstitial phenotype. The transcriptome of these macrophages shows enrichment of genes associated with prostaglandin biosynthesis and genes that drive T cell chemotaxis and differentiation. Therapeutic depletion of post-resolution macrophages, inhibition of prostaglandin E2 (PGE$_2$) synthesis or treatment with an EP4 antagonist, MF498, reduce numbers of lung CD4$^+$/CD44$^+$/CD62L$^+$ and CD4$^+$/CD44$^+$/CD62L$^-$/CD27$^+$ T cells as well as their expression of the α-integrin, CD103. The T cells fail to reappear and reactivate upon secondary challenge for up to six weeks following primary infection. Concomitantly, EP4 antagonism through MF498 causes accumulation of lung macrophages and marked tissue fibrosis. Our study thus shows that PGE$_2$ signalling, predominantly via EP4, plays an important role during the second wave of immune activity following resolution of inflammation. This secondary immune activation drives local tissue-resident T cell development while limiting tissue injury

Initiation and resolution of immune responses to infection are actively controlled processes. Resolution clears inflammatory mediators, promotes efferocytosis of apoptotic cells[1,2] and prevents chronic inflammatory diseases from developing[3]. However, we[4–6] and others[7–10] have recently described how resolution does not return tissues to homoeostasis as previously believed. Instead, resolution is followed by a phase of prolonged immune activity, which induces sentinel adaptive lymphocytic responses, maintains tolerance and invokes local tissue immunity[6,10].

For instance, following clearance of influenza A virus, mice lungs are populated by monocyte-derived macrophages that acquire an alveolar phenotype, which are functionally different to resident alveolar macrophages. These monocyte-derived macrophages persist in the lungs for up to 2 months and provide improved protection from subsequent S. *pneumoniae* infection[9]. Similarly, enhanced lung protection after self-limiting S. *pneumoniae* respiratory infections arose from a remodelling of the alveolar macrophage pool that is long-lasting and results in a macrophage population more adept at

responding to secondary infection[7]. Indeed, these post-resolution macrophages arise from new recruitment plus training of both the original cells and the new recruits[8]. In contrast, after resolution of inflammation to primary pneumonia, murine alveolar macrophages exhibited poor phagocytic capacity for several weeks thereby potentially explaining susceptibility to secondary infections. Unlike the phagocytic-proficient macrophages that appeared after resolution of influenza virus infection, and that were distinct from resident alveolar macrophages[9], these immunocompromised alveolar macrophages developed from resident alveolar macrophages that underwent an epigenetic programme in a signal-regulatory protein alpha dependent manner[10].

Hence, new areas of host defence following resolving inflammation that shape long-term tissue immunity are being discovered. We originally found the re-infiltration of prostanoid-secreting macrophages alongside lymph node expansion and increased numbers of blood and peritoneal memory T and B lymphocytes following the resolution of acute peritonitis driven by low-dose zymosan[5,6]. As confirmed elsewhere[9,11] these monocyte-derived macrophages remain in tissues for months dictating the magnitude of subsequent acute inflammatory reactions[5]. In contrast, injecting high concentrations of zymosan, a model of multiple organ failure, resulted in impaired resolution including a lack of lipid biosynthesis, but the appearance of antibodies to dsDNA indicative of tolerance being broken. Rescuing this system with a stable $PGE_2$ analogue, to reflect events that occurred following low dose zymosan, reduced levels of these antibodies indicating that experimentally introducing signals found in resolving inflammation can reset immunity.

Extending these studies to a clinically relevant model, we infected mice with live *S. pneumoniae* and found that as inflammation resolved there was a re-infiltration of monocyte-derived macrophages bearing an alveolar or interstitial phenotype. Post-resolution alveolar macrophages are enriched with factors that sustain T cell differentiation including prostaglandins with EP4 being the predominant prostaglandin receptor expressed on both T cells and macrophages. It transpires that post-resolution $PGE_2$ acting *via* EP4 exerts a dual role by driving local tissue-resident T cell development on the one hand, whilst counter-regulating macrophage trafficking and limiting tissue injury and fibrosis on the other.

Thus, we identify an inflammatory resolution pathway that confers long-term tissue protection whilst limiting injury following primary infection. We propose that these findings may explain how despite *S. pneumoniae* typically causing robust local inflammation, there is little consequential long-term tissue damage. By corollary, factors that affect resolution including old age or those that block post-resolution $PGE_2$ including nonsteroidal ant-inflammatory drugs (NSAIDs) may unwittingly impair the development of post infection tissue immunity and predispose to collateral tissue damage.

## Results

### *S. pneumoniae* triggers a transient resolving inflammation
The resolution of the pulmonary inflammatory response to *S. pneumoniae* was investigated using a previously described model of resolving non-bacteraemic pneumonia with the serotype 19F strain EF3030. Inoculation of *S. pneumoniae* into mouse lungs caused a transient weight loss 24 h after infection, which recovered over 4/6 days, Fig. 1A. Analysis of the digested lung using polychromatic flow cytometry revealed a biphasic peak in total immune cell infiltration peaking initially at 1–4d post bacteria injection and then again at day 14, Fig. 1B. This first peak comprised neutrophils (Fig. 1C) whose infiltration coincides with pro-inflammatory TNFα and IL-6 as well as anti-inflammatory IL-10 synthesis (Fig. 1D). By day 14 post infection no live *S. pneumonia* were detected within the lung and neutrophils and raised TNFα, IL-6 and IL-10 were cleared, Fig. 1E. In contrast there was a biphasic expression of TFGβ with an early increase in expression

followed by a fall then another increase at day 14 (Fig. 1D). In parallel, we noted a biphasic infiltration of total T and B cells (Fig. 1F) as well as cells of the myeloid lineage peaking 24 h after infection and then again at day 14, Fig. 1G. Immunohistochemistry confirmed the initial infiltration and clearance of neutrophils (Fig. 1H) while confocal analysis confirmed the re-appearance of myeloid and CD3[+] lymphoid cells post-resolution, Fig. 1I.

Collectively, these data show that intranasal challenge with *S. pneumoniae* is a model of acute inflammation with subsequent resolution occurring within three to 4 days post inoculation that is followed by an infiltration of lymphocytes and cells of the mononuclear lineage.

### Resolving pneumonia precedes a second wave of mononuclear phagocyte and lymphocyte infiltration
Next, we examined the mononuclear phagocytes and T cell subtypes that infiltrated the lung post-resolution. Using the gating strategy in Fig. 2A and Supplementary Figs. 1 and 9, mononuclear phagocytes comprising Ly6c[lo] monocytes peaked at day 1 following *S. pneumoniae* challenge and reappeared again at days 4–14 Fig. 2B; macrophages bearing an interstitial phenotype were found at day 2 and then again between days 7–14, Fig. 2C. Ly6c[hi] monocytes peaked at days 1–2 and day 14 (Fig. 2D), a profile that mirrored the infiltration of macrophages bearing an alveolar phenotype, which peaked at days 4 and 14, Fig. 2E. Recent research has revealed at least three sub-populations of interstitial macrophage based upon expression of LYVE-1 and MHC-II. Use of antibodies against these cells surface markers showed that post-resolution interstitial macrophages comprise LYVE-1[hi]/MHC-II[neg]; LYVE-1[neg]/MHC-II[hi] and double negative LYVE-1[neg]/MHC-II[neg] cells. The most pronounced population was the second peak of primarily LYVE-1[hi]/MHC-II[neg] interstitial macrophages at day 14, Fig. 2F–H.

Using the gating strategy in Supplementary Fig. 1 to describe T cell profiles, we found CD4[+]/CD44[+]/CD62L[+] T cells, consistent with a central memory T cell phenotype, peak at day 14. Additionally, these central memory-like cells were enriched for CD103 at day 14 with expression of CD103 on CD4[+]/CD44[+]/CD62L[-]/CD27[+]/CD103[+] early effector memory like T cells peaking 6 weeks later. Day 14 lymphocytes produced high levels of IFNγ, IL-17 and TNFα. Lymphocytes producing cytokines without ex vivo re-stimulation comprised mostly of CD44[+]/CD62L[-]/CD27[-]/CD103[-] lymphocytes, Supplementary Fig. 2.

### Transcriptomic analysis of post-resolution macrophages
To understand post-resolution mononuclear phagocyte function and the immuno-biology they control, macrophages bearing an alveolar phenotype as well as the three populations of interstitial macrophages (Fig. 2E–H) were sorted using fluorescence-activated cell sorting (FACS) from naïve as well as day 14 post *S. pneumonia* challenged mice and their transcriptomes sequenced using Illumina NovaSeq. Using a fold enrichment of logFC = ± 3 and $P < 0.05$ no significant GO-terms with PANTHER or GoRilla or pathways with Reactome were detected in day 14 versus naïve LYVE-1[hi]/MHC-II[neg]; LYVE-1[neg]/MHC-II[hi] populations. However, a list of top gene changes as per fold enrichment are included in Fig. 3A–C. Of note, comparing the LYVE-1[neg]/MHC-II[hi] cells both *Saa3* and *Iigp1* were upregulated at day 14 compared to naïve.

In contrast, alveolar macrophages showed a significant enrichment of gene expression profiles with 33 genes significantly upregulated in day 14 alveolar macrophages compared to alveolar macrophages from naïve lungs, FDR < 0.05, Fig. 3D. Of these unbiasedly assigned GO terms, the most abundant centred around lymphocyte chemotaxis and positive regulation of T cell migration (CXCL10, CXCL16; FDR = 0.04, FDR = 0.002), factors that positively regulate T cell differentiation (InhBa, IL-7r; FDR = $3.5 \times 10^{-7}$, FDR = 0.01) as well as factors that control mononuclear cell migration (CCL5, 7 and their receptors CCR5, 11; $P = 0.006$, $P = 0.007$ and $P = 7 \times 10^{-4}$, $P = 2 \times 10^{-5}$); mononuclear phagocyte response to pathogens and products of bacterial origin (PTGS2 [COX-2], CxCL1, TNFAIP3 and

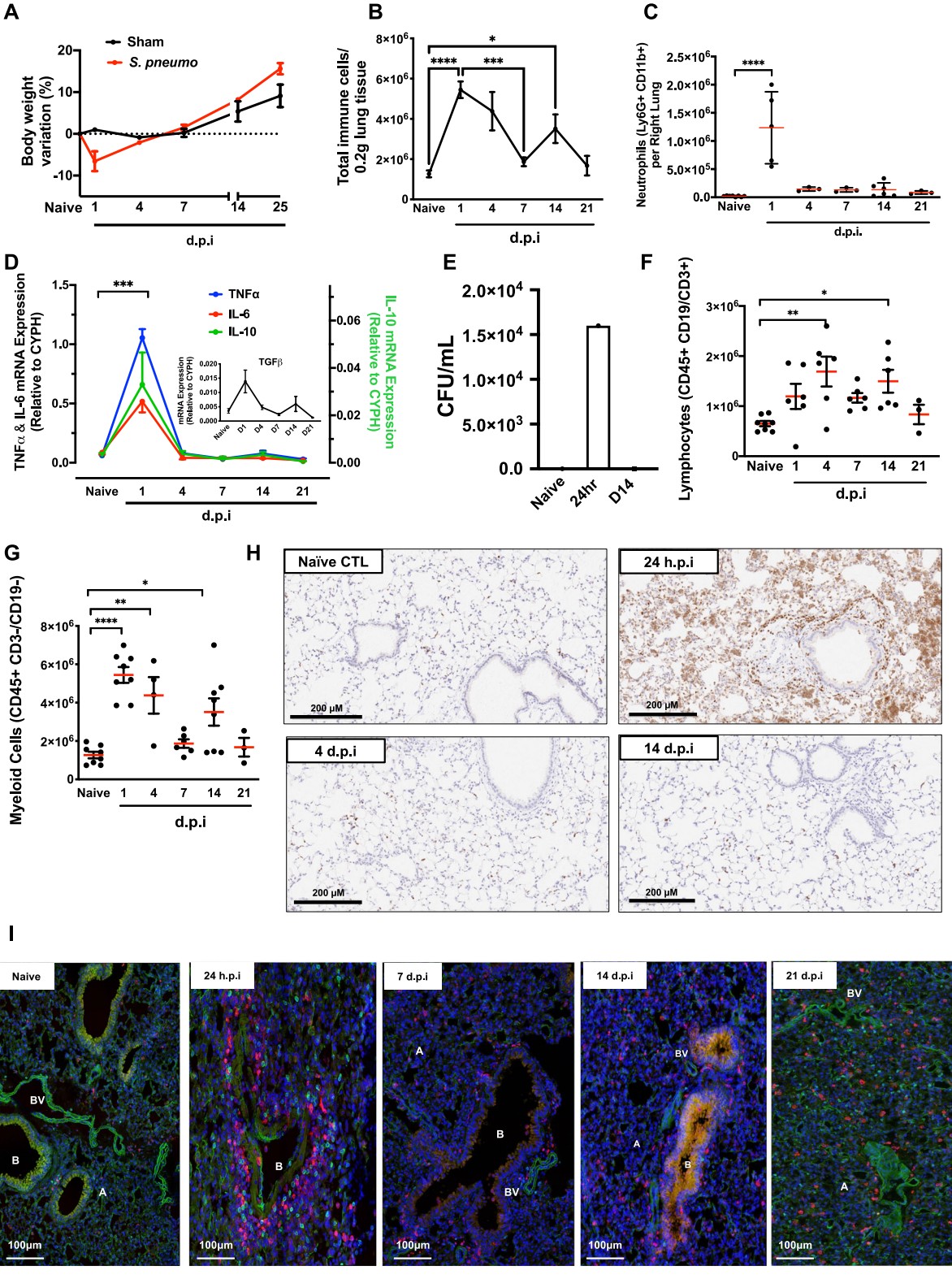

**Fig. 1 | Intranasal *S. pneumoniae* causes a self-resolving inflammation.** WT C57BL6/J mice were administered intranasal *S. pneumoniae* with animals (**A**) experiencing a transient weight loss. Lungs were digested and samples analysed by polychromatic flow cytometry for markers of acute inflammation including (**B**) total leucocytes, (**C**) neutrophils, (**D**) pro-inflammatory cytokines and (**E**) bacterial clearance (Colony Forming Units/mL of bronchoalveolar lavage fluid). Flow cytometry was also used to profile (**F**) total lymphoid and (**G**) myeloid cells throughout inflammation, resolution and weeks following resolution. The temporal profile of (**H**) neutrophils (GR1+) as well as (**I**) macrophages (F4/80+) and CD3 T cells (CD3+) were confirmed at tissue level, panels H-I. A Alveoli, B bronchiole, BV blood vessel. Sections are representative of $n = 3$ independent experiments. Data were analysed by one-way analysis of variance (ANOVA) and Tukey's multiple comparisons test. A $p$ value of <0.05 was taken as the threshold of significance with graphical representation as; $p < 0.05 = *$, $p < 0.01 = **$ and $p < 0.001 = ***$ and presented as mean ± SEM. ($n = 5$–8 mice/group).

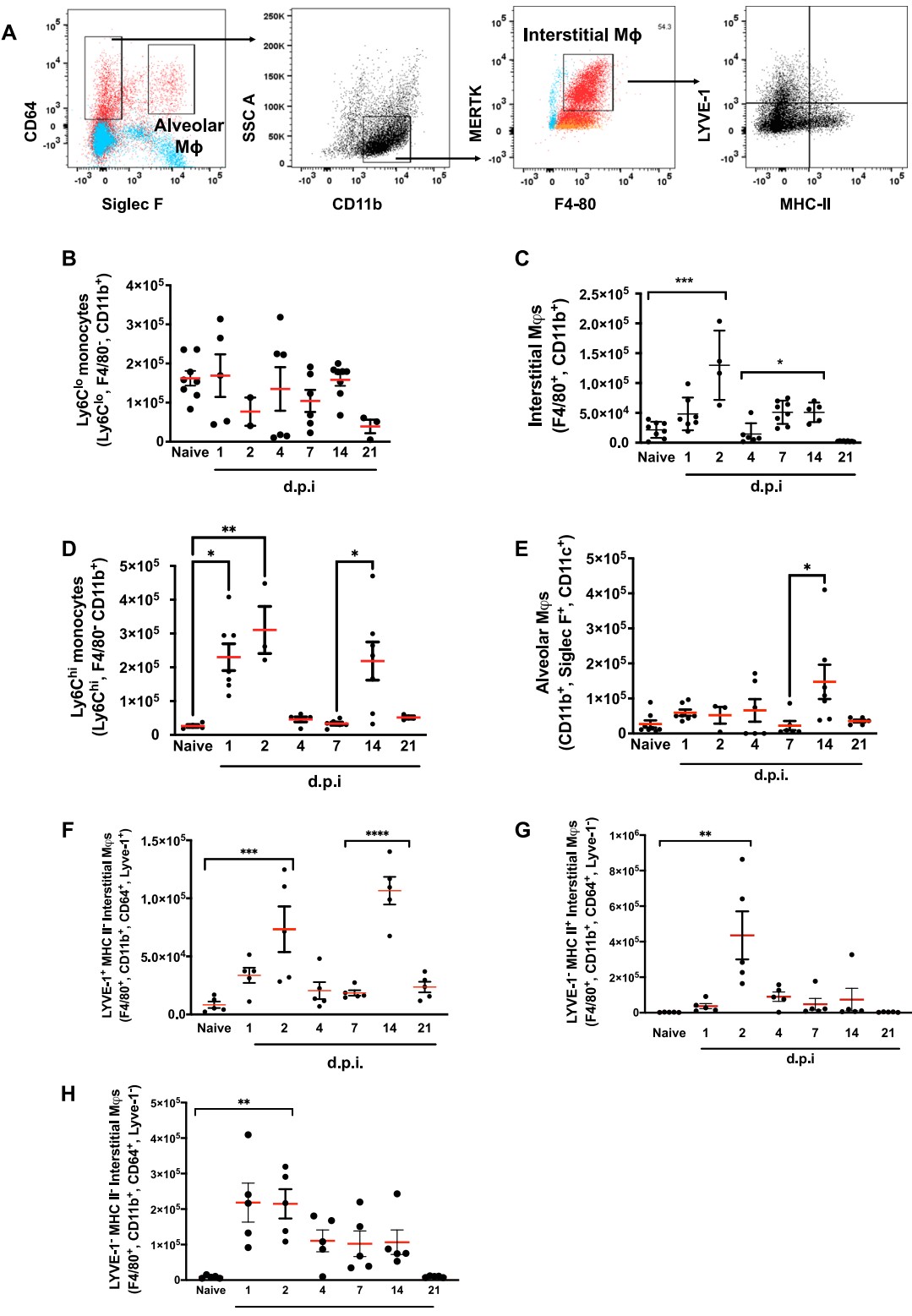

**Fig. 2 | Resolution of lung inflammation is followed by a second wave of myeloid cell infiltration.** WT C57BL6/J mice were administered intranasal *S. pneumoniae* with lungs digested for analysis by polychromatic flow cytometry. Using the gating strategy in panel (**A**) (FMOs are shown in blue to identify positive populations (red)) profiles of (**B**–**E**) monocytes, interstitial and alveolar macrophage population were identified with panels (**F**–**H**) providing further analysis in interstitial macrophage sub-types. Data were analysed by one-way analysis of variance (ANOVA) and Tukey's multiple comparisons test. A $p$ value of <0.05 was taken as the threshold of significance with graphical representation as; $p < 0.05 = *$, $p < 0.01 = **$ and $p < 0.001 = ***$ and presented as mean ± SEM ($n = 3$–8 mice/group).

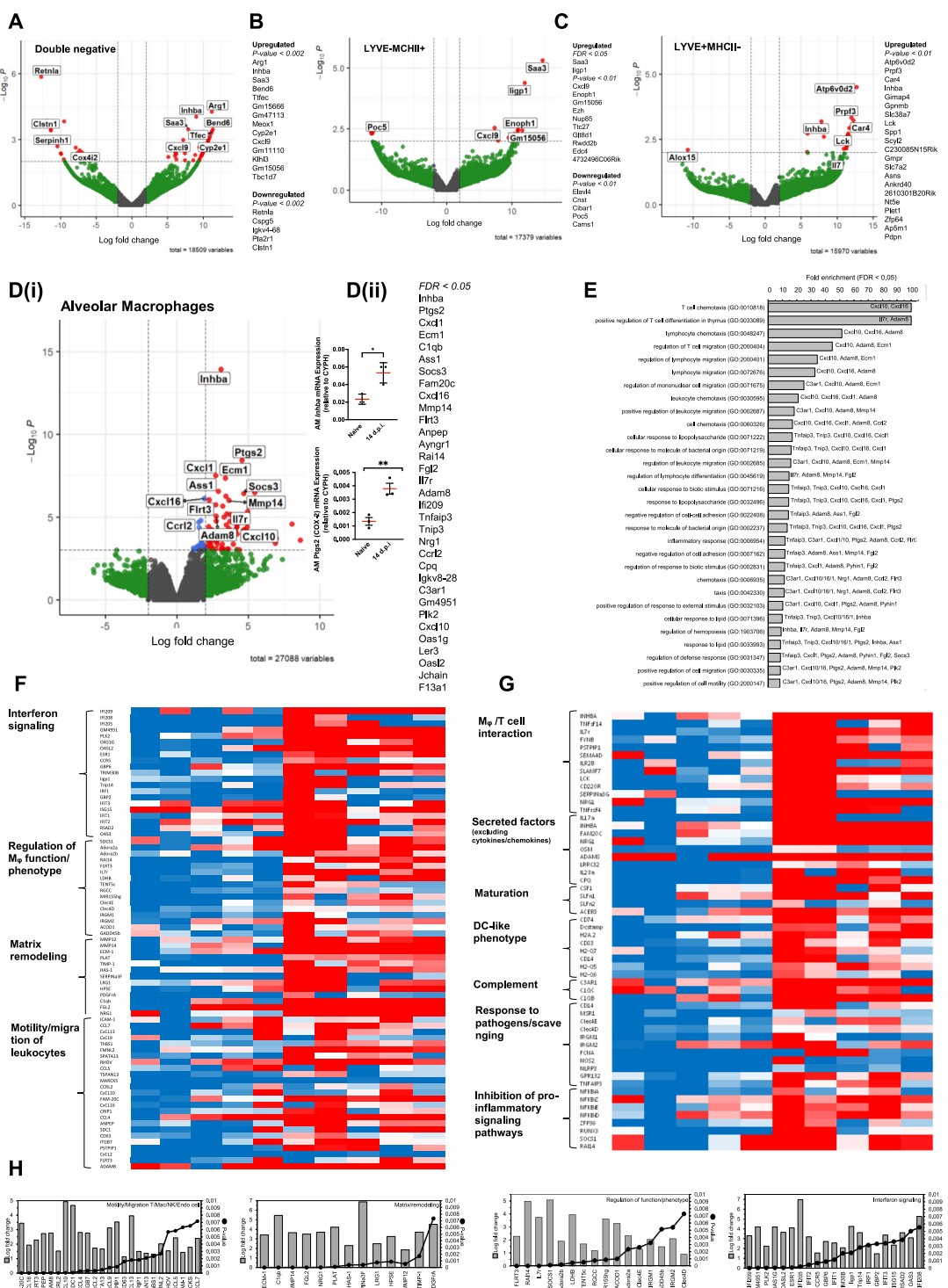

**Fig. 3 | RNAseq analysis reveals a role for post-resolution macrophage populations in T cell migration/maturation.** WT C57BL6/J mice were administered intranasal *S. pneumoniae* with lungs digested and macrophage populations including (**A**) double negative interstitial macrophage, (**B**) LYVE-1⁻/MHC-II⁺ interstitial macrophages, (**C**) LYVE-1⁺/MHC-II⁻ interstitial macrophages as well as (**D**) alveolar macrophages sorted by FACS and subject to analysis by RNAseq followed by bioinformatic analysis using edgeR in RStudio as illustrated (*n* = 5 mice/group). **D** qPCR validation (*n* = 3 mice/group) was used in a separate independent experiment to confirm upregulation of *Inhba* and *Ptgs2* in naïve versus day 14 alveolar macrophages. Students unpaired *t*-test was used to compare the means of two groups. A *p* value of <0.05 was taken as the threshold of significance with graphical representation as; *p* < 0.05 = *, *p* < 0.01 = ** and *p* < 0.001 = *** and presented as mean ± SEM. As alveolar macrophages showed the greatest changes post-resolution compared to the naive state these cells were further analysed by using (**E**) PANTHER to identify GoTerms from upregulated genes at day 14 compared to naïve, (**F**, **G**) normalised read counts of five replicates from naïve alveolar macrophages compared to day 14, (**H**) EdgeR results showing differential gene expression as logFC and *p* value for genes relating to migration, matrix remodelling, regulation of phenotype and interferon signalling.

TNIP3; FDR = $5 \times 10^{-5}$, FDR = $3 \times 10^{-4}$, FDR = 0.02, FDR = 0.01) were also enriched in day 14 alveolar macrophages. Of note was IL7r, which is reported to have a logFC = 3.7 enrichment on alveolar macrophages at day 14 compared to naïve. Modulation of cell-cell adhesion (ADAM8, ASS1, FGL2; FDR = 0.01, FDR = 0.001, FDR = 0.01) and responses to lipids (TNFAIP3, TNIP3, CXCL10, CXCL16, CXCL1 and PTGS2) were also reported, Fig. 3E.

However, a closer examination of this list revealed genes involved with the negative regulation of proinflammatory signalling pathways (TNFAIP3, SOCS FDR = 0.001), matrix remodelling (MMP14 FDR = 0.002, ECM-1 FDR = $3 \times 10^{-4}$, FGL2) and interferon signalling (PLK2, FDR = 0.04), for instance, that were not highlighted by GO term search. This led us to draft a list of ~200 genes within FDR 0.01. Manual curation of these genes revealed ~11 functional categories ranging from T cell and mononuclear cell migration and T cell differentiation as highlighted by unbiased GO term analysis as above, but also a more extensive list of interferon signalling molecules, factors that regulate macrophage phenotype, maturation/differentiation and function, matrix remodelling and non-cytokine/chemokine secretory factors involved in immune regulation, Fig. 3F–H. Figure 3F, G represent normalised read numbers from five replicates of alveolar macrophages from the naïve lung to day 14. However, Fig. 3H represents average fold changes between these same samples and details the statistical significance in differential expression of chosen genes. All differentially expressed genes were upregulated at day 14 compared to naïve, there were not any genes statistically significantly downregulated.

Taken together, while interstitial macrophages underwent some changes post-resolution compared to the naïve state, alveolar macrophages experienced distinct changes in phagocyte and lymphocyte migration, control of macrophage and T cell differentiation, matrix remodelling as well as negative regulation of pro-inflammatory signalling pathways and priming for responses to secondary viral and bacterial infection.

## Origin of post-resolution macrophages
To investigate if the second wave of post-resolution mononuclear phagocytes is dependent on circulating monocytes, an anti-CCR2 antibody, MC-21, was utilised to deplete CCR2$^+$ monocytes in the peripheral blood. Dosing mice with MC-21 at days 6, 8 and 10 post *S. pneumoniae* challenge (Fig. 4A) reduced numbers of blood Ly6c$^{hi}$ as well as Ly6c$^{lo}$ monocytes (Supplementary Fig. 3) resulting in a significant reduction in lung monocyte populations at day 14 (Fig. 4B, C). Consequently, MC-21 reduced numbers of alveolar macrophages (Fig. 4D) but completely blocked the appearance of LYVE-1$^{hi}$/MHC-II$^{neg}$ interstitial macrophages, Fig. 4E.

Analysis of the lymphocyte compartment in the lung after depletion of day 14 macrophage population revealed that while CD4$^+$/CD62L$^+$/CD44$^-$ naïve T cell numbers were unaffected, there was a significant reduction in CD4$^+$/CD44$^+$/CD62L$^+$ T cells as well as populations of these cells expressing CD103 Fig. 4H–J, an α-integrin that maintains residency of T cells within tissues. We repeated these experiments by administering MC-21 from day 6 to day 10 post-resolution after which MC-21 was withdrawn with lungs examined at 6 weeks. At this later time point numbers of T cells bearing an early effector memory T cell phenotype (CD4$^+$/CD44$^+$/CD62L$^-$/CD27$^+$) and a sub-population of these cells expressing CD103 remained at baseline levels, Fig. 4K–L. These data suggest that all post-resolution interstitial and a proportion of alveolar macrophages are monocyte-derived and, in line with predictions from transcriptomic analysis from Fig. 3, these cells are required for the appearance of differentiated T cells in the lung.

## Post-resolution prostanoid biosynthesis
To identify the endogenous factors that control post-resolution macrophage infiltration and T cell function, we noted that day 14 alveolar macrophages were enriched with PTGS2 (COX-2), Fig. 3D. Indeed, prostanoids have well established diverse effects on mononuclear phagocyte and T cell phenotypes[12–15]. Hence, given our previous work where post-resolution zymosan-induced peritonitis saw a robust and sustained synthesis of prostaglandins within the cell-free inflammatory exudate[6], we examined whether a similar phenomenon existed following the resolution of lung pneumonia. LC-MS/MS analysis found that following its expected early peak at onset, there was a second wave of PGE$_2$ peaking again at days 7–14 (Fig. 5A, B); the full profile of lipids as determined by LC-MS/MS are displayed in Supplementary Data 1. Mononuclear phagocytes sorted by FACS from naïve and post-resolution lungs revealed that post-resolution alveolar macrophages were enriched with COX-2 (Fig. 5C) and inducible downstream prostaglandin E synthase, mPGES-1, was upregulated in both post-resolution alveolar and interstitial macrophages, Fig. 5D.

We also investigated the expression profile of the four PGE$_2$ receptors, EP1-4, in lung macrophages and lymphocytes. EP4 was the most predominantly expressed receptor on post-resolution macrophages (Fig. 5E, F), with EP4 also the predominant PGE$_2$ receptor expressed on T cells, Fig. 5G. We also found a second peak in prostacyclin (PGI$_2$) synthesis post-resolution (Fig. 5H, I), most likely arising from infiltrating monocytes, Fig. 5J.

In addition to mRNA and prostanoid profiles, we also report prostanoid biosynthetic machinery and receptor expression at the protein level during inflammation and post-resolution. For instance, there is co-expression of COX-2 and mPGES-1 within macrophages and stromal cells 24 h after intranasal *S. pneumoniae*, as well as post-resolution, Supplementary Fig. 4A–D. In addition, while both EP2 and EP4 were expressed at 24 h post *S. pneumonia* infection, EP4 predominated at day 14, Supplementary Fig. 5A–D. Indeed, with COX-2 and mPGES-1 being expressed in macrophages post-resolution, MC-21 caused a significant reduction in PGE$_2$ (Supplementary Fig. 6) in line with the depletion of these cells, Fig. 4.

Taken together these data show a robust and sustained wave of prostaglandin biosynthesis after inflammatory resolution, with PGE$_2$ and EP4 being the predominant EP receptor expressed on lymphocytes and mononuclear phagocytes post-resolution.

## Post-resolution PGE$_2$ controls T lymphocyte populations
To investigate a potential link between post-resolution macrophage-derived PGE$_2$ and EP4 with T cell numbers and phenotypes, minipumps loaded with the pan-COX inhibitor naproxen or the EP4 antagonist MF498 were implanted into mice following resolution of their lung inflammation (day 4) and the composition of the lung examined 6 weeks after initial infection, Fig. 6A. Neither naproxen nor MF498 affected numbers of CD4$^+$/CD44$^+$/CD62L$^-$/CD27$^-$ late effector memory T cells (TeML, Fig. 6B) but reduced their secretion of IL-17 and IFNγ, Fig. 6C. In contrast, blocking PGE$_2$ synthesis or antagonising EP4 reduced numbers of CD4$^+$/CD44$^+$/CD62L$^-$/CD27$^+$ early effector memory T cells (TeME, Fig. 6D) as well as numbers of these cells expressing CD103, Fig. 6E. These data resemble the effects of eliminating post-resolution macrophages on TeM numbers and phenotype, Fig. 4. Indeed, EP4 antagonism reduced CD103 expression levels to below those seen in naive T cells, Fig. 6F. To determine whether this is a direct effect of PGE$_2$ on tissue T cells, we isolated splenocytes from naïve mice and treated them with PGE$_2$. PGE$_2$ did not elevate CD103 directly, but antagonising endogenous PGE$_2$ significantly dampened CD103 expression, Fig. 6G. TGFβ is known to drive T cell differentiation[16] and was shown to be co-elevated with post-resolution lipids in this model of lung inflammation (Fig. 1D and Fig. 6H). The combination of TGFβ and PGE$_2$ and synergistically elevated IL-17 and IFNγ release from T cells in an EP4 dependent manner, Fig. 6I–J. These data suggest that post-resolution PGE$_2$ signalling via EP4 helps to generate CD4$^+$/CD44$^+$/CD62L$^-$/CD27$^+$ and CD4$^+$/CD44$^+$/CD62L$^-$/CD27$^+$/CD103+ resident (TrM) T cells.

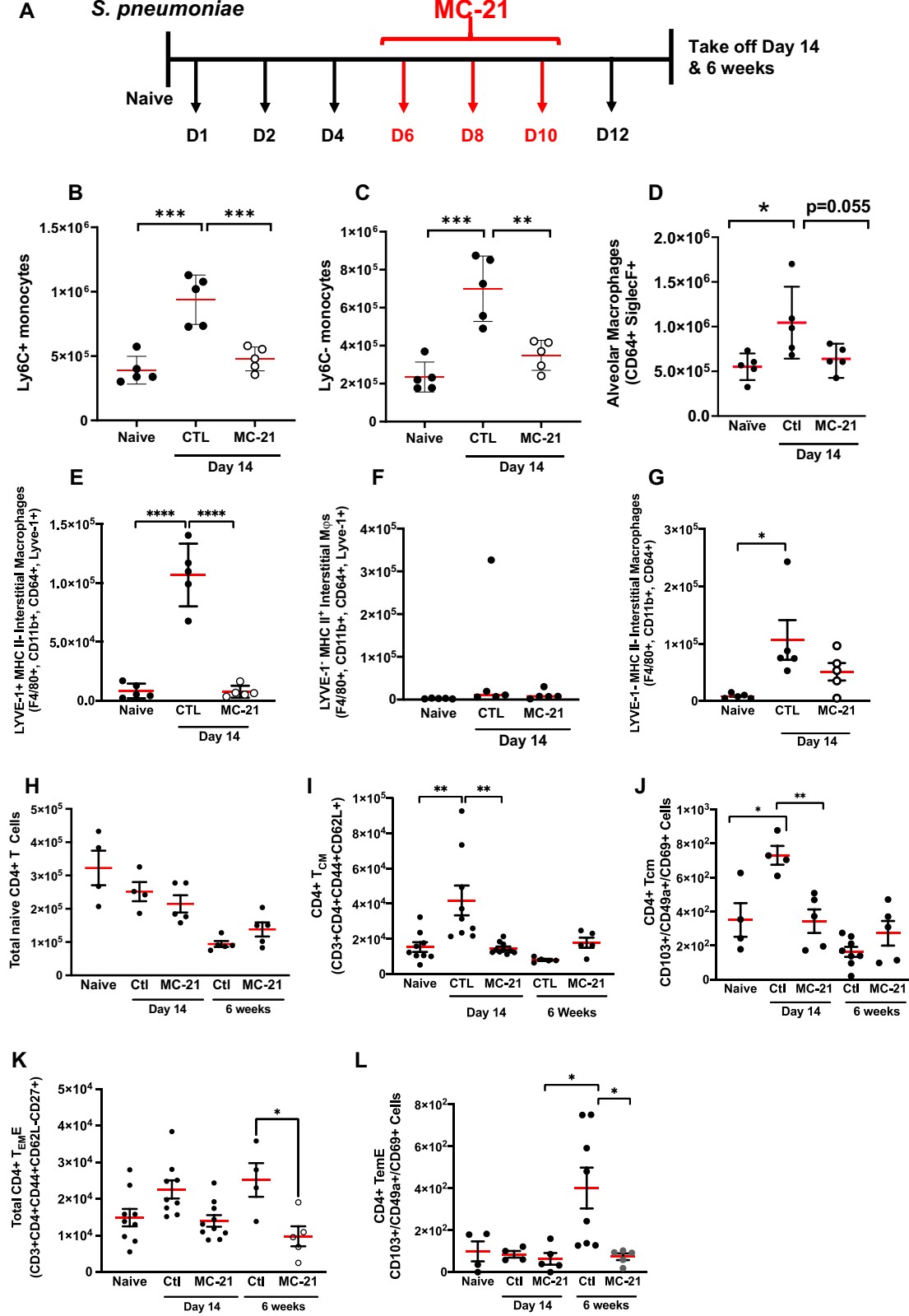

**Fig. 4 | Therapeutic deletion of blood monocyte reveals the source of post-resolution macrophages and a role for these cells in T cell maturation.** MC-21, which depletes blood monocytes was administered therapeutically (**A**) six days after intranasal *S. pneumoniae* and every second day up to day 14. Polychromatic flow cytometry was carried out for profiles of lung (**B**–**D**) monocytes and macrophages as well as (**E**–**G**) interstitial macrophage sub-populations. In addition, we determined the impact of depleting post-resolution macrophages on profiles of lung tissue (**H**–**L**) CD4$^+$ T cell subpopulations. Data were analysed by one-way analysis of variance (ANOVA) and Tukey's multiple comparisons test. A *p* value of <0.05 was taken as the threshold of significance with graphical representation as; $p < 0.05$ = *, $p < 0.01$ = ** and $p < 0.001$ = *** and presented as mean ± SEM ($n = 5$ mice/group).

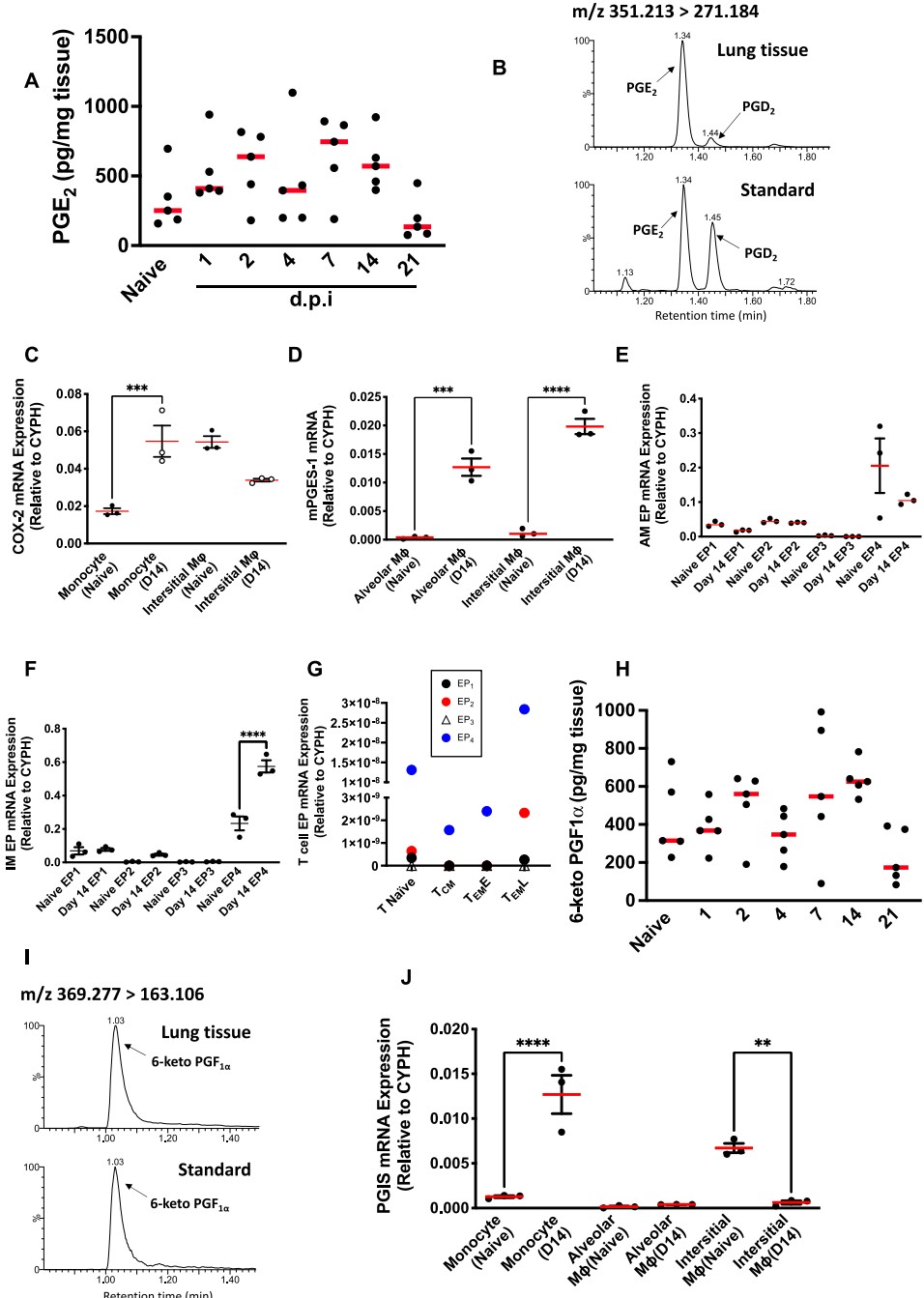

**Fig. 5 | Post-resolution prostaglandin synthesis regulates monocyte infiltration via EP4/CCL2.** Whole lungs were isolated at the indicated times following intranasal *S. pneumoniae* for lipid extraction and analysis by LC-MS/MS revealing the profile of (**A**) PGE$_2$ and (**B**) a representative reconstructed ion chromatogram. PCR analysis was carried out on (**C**, **D**) post-resolution macrophages to identify the source of PGE$_2$ as well as (**E**, **F**) the predominant PGE$_2$ receptor (EP) that is expressed on alveolar and interstitial macrophages, respectively, as well as post-resolution (**G**) T cell populations. A similar biphasic profile for prostacyclin (PGI$_2$), measured as its stable derivative 6-keto PGF1α, is presented in (**H**, **I**) along with (**J**) the post-resolution cells that synthesise it. Data were analysed by one-way analysis of variance (ANOVA) and Tukey's multiple comparisons test. A *p* value of <0.05 was taken as the threshold of significance with graphical representation as; $p < 0.05 = *$, $p < 0.01 = **$ and $p < 0.001 = ***$ and presented as mean ± SEM ($n = 3$–5 mice/group).

## Post-resolution PGE$_2$ via EP4 determines the re-emergence of T cells

Following the above, we counter-stimulated these animals at week 6 post initial infection with a second challenge of *S. pneumoniae* and examined events that occurred 24 h later, Fig. 7A. As expected, secondary challenge with *S. pneumoniae* caused an increase in numbers of CD4$^+$/CD44$^+$/CD62L$^+$ cells with both naproxen and MF498 reducing these back to control levels, Fig. 7B. Upon activation, TrM within the

lung lose CD103 to facilitate their movement within infected tissues. While it is not entirely clear why TrM downregulate their residency programme, recent studies demonstrate that upon antigen exposure TrM can egress and traffic to other tissues conferring protection beyond their local environment[17,18]. This might explain why we found reduced numbers of CD4$^+$/CD44$^+$/CD62L$^-$/CD27$^+$ cells in lungs 24 h following re-challenge with *S. pneumoniae*, Fig. 7C (ctl Vs 6 w.p.i). Indeed, we found that CD103 was down regulated on remaining CD4$^+$/

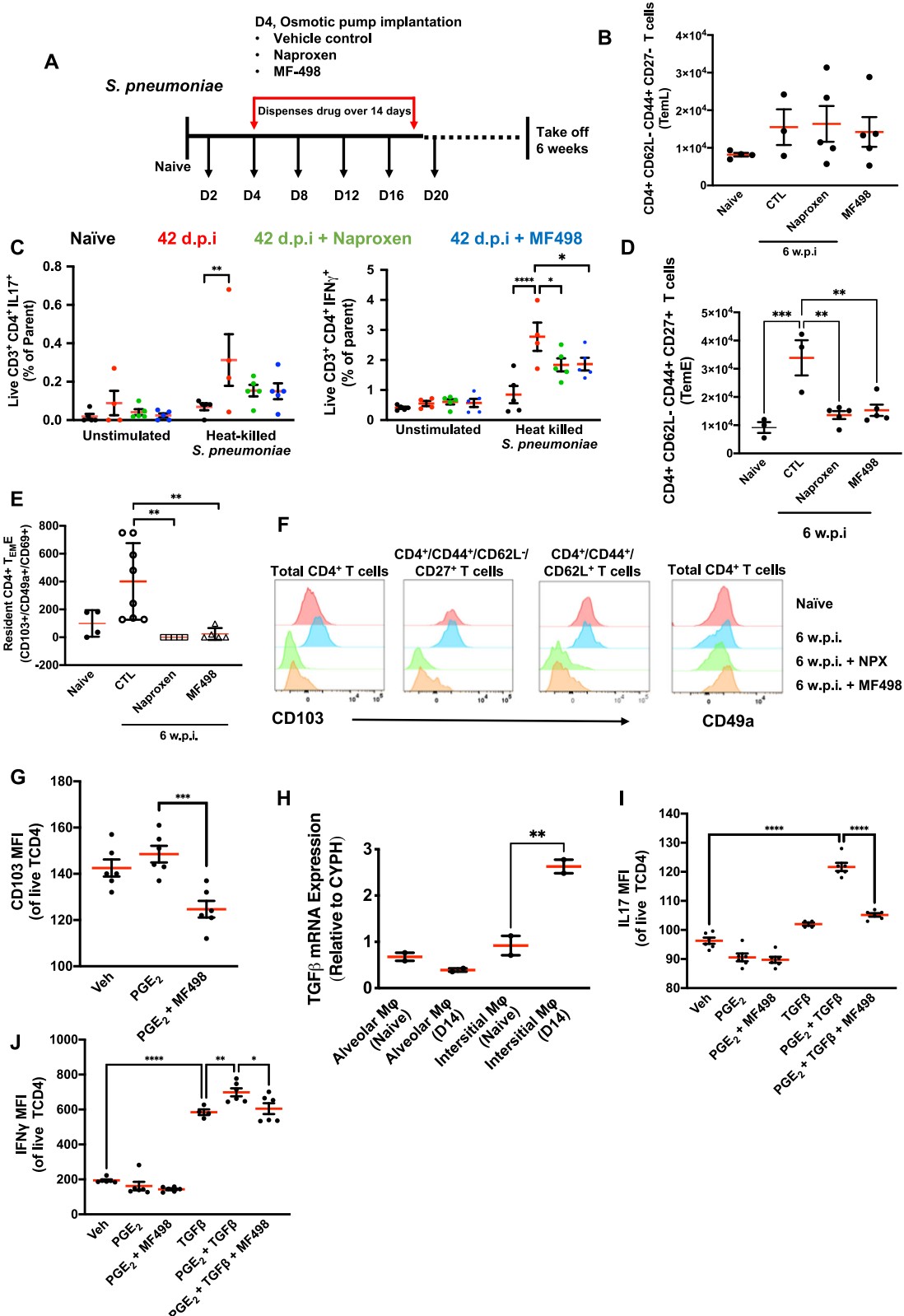

CD62L⁻/CD44⁺/CD27⁺ cells upon re-introduction of *S. pneumoniae* in a manner that was blocked by MF498 and naproxen, Fig. 7D. MF498 and naproxen also blocked expression of the activation markers CD69 and CD44, Fig. 7E, F. However, despite effector T cell and PMN numbers (Fig. 7G) being reduced in the lungs of rechallenged animals treated with a PGE₂ inhibitor or EP4 antagonism, live *S. pneumoniae* was efficiently cleared in both treatment groups, Fig. 7H.

## Blocking post-resolution PGE₂ via EP4 causes macrophage infiltration and tissue fibrosis

To understand this paradox, we noted that transiently blocking post-resolution EP4 by implanting minipumps loaded with MF498 from day 4 and examining lungs at day 14 (Fig. 8A), revealed a marked tissue fibrosis with a mean Ashcroft fibrosis score of 5.5 (SD ± 0.5, $n = 3$) versus vehicle control at day 14 of 0 (Fig. 8B). Indeed, Supplementary

**Fig. 6 | PGE$_2$ via EP4 controls post-resolution T cells populations and phenotype. A** Osmotic pumps loaded with either the pan COX inhibitor naproxen or the EP4 antagonist MF498 were implanted into mice once inflammation resolved (day 4) and their effects of blocking the biological action of post-resolution PGE$_2$ was established 6 weeks later as determined by measuring (**B**) late effector T cell numbers and (**C**) their effector function as well as (**D**) early effector T cells and their expression of (**E, F**) the α-integrin, CD103. as a marker of T cell residence potential. Linking PGE$_2$ with this post-resolution T cell phenotype, T cells were incubated PGE$_2$ and its direct effect on (**G**) CD103 expression or with other cytokines expressed during post-resolution that are known to affect lymphocyte function, (**H**) namely TGFβ, were examined for their collective effects on intracellular (**I**) IL-17 and (**J**) IFNγ. Data were analysed by one-way analysis of variance (ANOVA) and Tukey's multiple comparisons test. A $p$ value of <0.05 was taken as the threshold of significance with graphical representation as; $p < 0.05$ = *, $p < 0.01$ = ** and $p < 0.001$ = *** and presented as mean ± SEM ($n = 3$–6 mice/group).

Fig. 7 shows transient tissue fibrosis in this *S. pneumoniae* model starting with a naïve baseline lung fibrosis Ashcroft score of 0 peaking at day 2 and resolving back to baseline by days 14/21. As expected, depleting post-resolution macrophages with MC-21 also resulted in increased fibrosis with a mean Ashcroft fibrosis score of 4.8 (SD ± 0.30, $n = 3$) versus vehicle control at day 14 of 0, Supplementary Fig. 8. In addition to fibrosis, we also noted a mononuclear cell infiltrate, Fig. 8C. This monocular cell infiltrate persisted with numbers of both alveolar and LYVE-1$^+$/MHC II$^-$ IMs detectable in the lung 6 weeks later, Fig. 8D–G. Macrophages that persisted up to 6 weeks following EP4 antagonism had a higher phagocytic capacity compared to macrophages from time-matched control animals, Fig. 8H.

To understand how PGE$_2$ acting via EP4 controls post-resolution macrophage infiltration we found whole lung expression of the classic monocyte chemoattractant CCL2 rising early in the response (day 1) and then again from day 4 (Fig. 8I) primarily enriched in IMs (Fig. 8J) with its receptor, CCR2, also enriched on these cells post-resolution, Fig. 8K–L. Isolating post-resolution macrophages and treating them ex vivo with a stable analogue of PGE$_2$ dampened CCL2 production in a manner that was reversed using the EP4 antagonist MF498, Fig. 8M.

Hence, these data suggest that the enhanced clearance of bacteria from secondary infected mice (Fig. 7L) arose from the increased numbers of compensatory macrophages that persisted in the lung tissue at week 6 following MF498 that are efficient at killing bacteria. However, this elevation in macrophages was also associated with increased tissue damage likely arising from the proinflammatory phenotype they acquired from post-resolution EP4 blockade.

## Discussion

We report that following the resolution of an inflammatory response to *S. pneumoniae* infection in the mouse lung, there is a sequence of events driven by macrophage derived PGE$_2$ that not only builds tissue immunity through the differentiation of T lymphocytes, but also limits tissue damage post-infection. Specifically, as soon as inflammation resolves PGE$_2$, working through EP4, brings about the differentiation of TcM and TrM cells such that antagonising EP4 or PGE$_2$ synthesis resulted in reduced numbers of these cells 14 days and 6 weeks following infection. More importantly, this intervention also caused failure of these cells to reappear upon re-stimulation with the same pathogen. Blocking EP4 during post-resolution also abrogated the expression of activation markers including CD103 on effector memory T cells bearing a tissue-resident phenotype. Concomitantly, PGE$_2$ via EP4 controls macrophage infiltration and limits tissue fibrosis. Taken together, we show there is a window of immune activity following resolving inflammation that limits tissue damage and shapes tissue immunity by creating an environment for the generation of tissue-resident memory T cells.

Therapeutically depleting post-resolution macrophages, blocking their secretion of PGE$_2$ or antagonising the predominant PGE$_2$ receptor expressed on T cells, EP4, all resulted in a reduction in CD4$^+$/CD44$^+$/CD62L$^-$/CD27$^+$ early effector memory T cells and their expression of CD103 for up to six weeks post-infection. Moreover, while number of late effector CD4$^+$/CD44$^+$/CD62L$^-$/CD27$^-$ T cells were not affected by this intervention, their secretion of IL-17 cells was inhibited. Based on the current model for the development of T cells and the data presented here, we propose that CD4$^+$/CD44$^+$/CD62L$^-$/CD27$^+$ early effector memory cells have two differentiation pathways whereby they may lose CD27 and become late effector memory like T cells or they gain CD103 and acquire tissue resident status; CD103 interacts with E-cadherin expressed on epithelial cells maintaining these cells in tissues. As treatment of T cells with the EP4 receptor antagonist MF498 significantly dampened the expression of CD103 both in vivo and in vitro, this suggests that the PGE$_2$/EP4 axis is required for the generation of tissue resident memory T cells following the resolution of acute inflammation as well as the development of effectors function of late effector T cells. Indeed, the immune ingredients for the development of the effector function of these cells were found during post-resolution including TGFβ which when co-incubated with PGE$_2$ and T cells in vitro drove IFNγ and IL-17 synthesis. Moreover, blocking post-resolution PGE$_2$ synthesis or antagonism of EP4 impaired release of these effector molecules from TrM in an antigen-dependent manner. In addition to CD103 and T cell activation markers including CD69, this adds to the repertoire of roles post-resolution PGE$_2$ acting via EP4 has in driving the development of long-term TrM cells.

Restimulating mice with *S. pneumoniae* 6 weeks after primary infection with *S. pneumoniae* resulted, as expected, an increase in lung CD4$^+$/CD44$^+$/CD62L$^+$ TcM 24 h later, but reduced numbers of CD4$^+$/CD44$^+$/CD62L$^-$/CD27$^+$ effector T cells and their expression of CD103. While it's unclear why TrM downregulate aspects of their residency programme, recent studies demonstrate that upon antigen exposure TrM can egress and traffic to other tissues conferring protection beyond their local environment[17,18]. Repeating these experiments and antagonising post-resolution EP4 (whilst introducing *pneumoniae* 6 weeks later) also caused significantly fewer CD4$^+$/CD44$^+$/CD62L$^-$/CD27$^+$ early effector T cells expressing CD103 compared to controls. This likely arose from the fact that there were fewer of these cells in the lung because of inhibiting post-resolution PGE$_2$ synthesis or EP4 antagonism, see Fig. 6C, D. Of interest, a look at these remaining cells revealed that they were unable to lower their expression of CD103 or upregulate activation marker CD69 upon seeing *S. pneumoniae* for the second time. These data suggest that following inflammatory resolution, while PGE$_2$ drives the development, at least in part, of a resident T cell phenotype, it also confers some long-term ability to re-activate following recognition of the same stimulus for optimal immune surveillance.

Despite our data showing that post-resolution PGE$_2$ imprints tissue immunity, the secondary infection was efficiently cleared by the host. It transpires that PGE$_2$ has a dual role in controlling post-resolution biology in that it also negatively regulates the infiltration of mononuclear phagocytes following resolution through inhibition of CCL2 expression. Consequently, blocking PGE$_2$ resulted the accumulation of alveolar macrophages and interstitial macrophages in the lung tissue that persisted for up to 6 weeks with these cells having increased bacterial phagocytic capacity compared to controls. We suspect that this helps to explain efficient bacterial clearance despite a paralysed T cell compartment, a type of compensatory response. However, this came at the expense of significant tissue fibrosis. Whilst this could have resulted from a direct effect of pro-inflammatory macrophages, PGE$_2$ has a direct inhibitory effects on fibroblast proliferation, migration, collagen secretion and fibroblast-to-

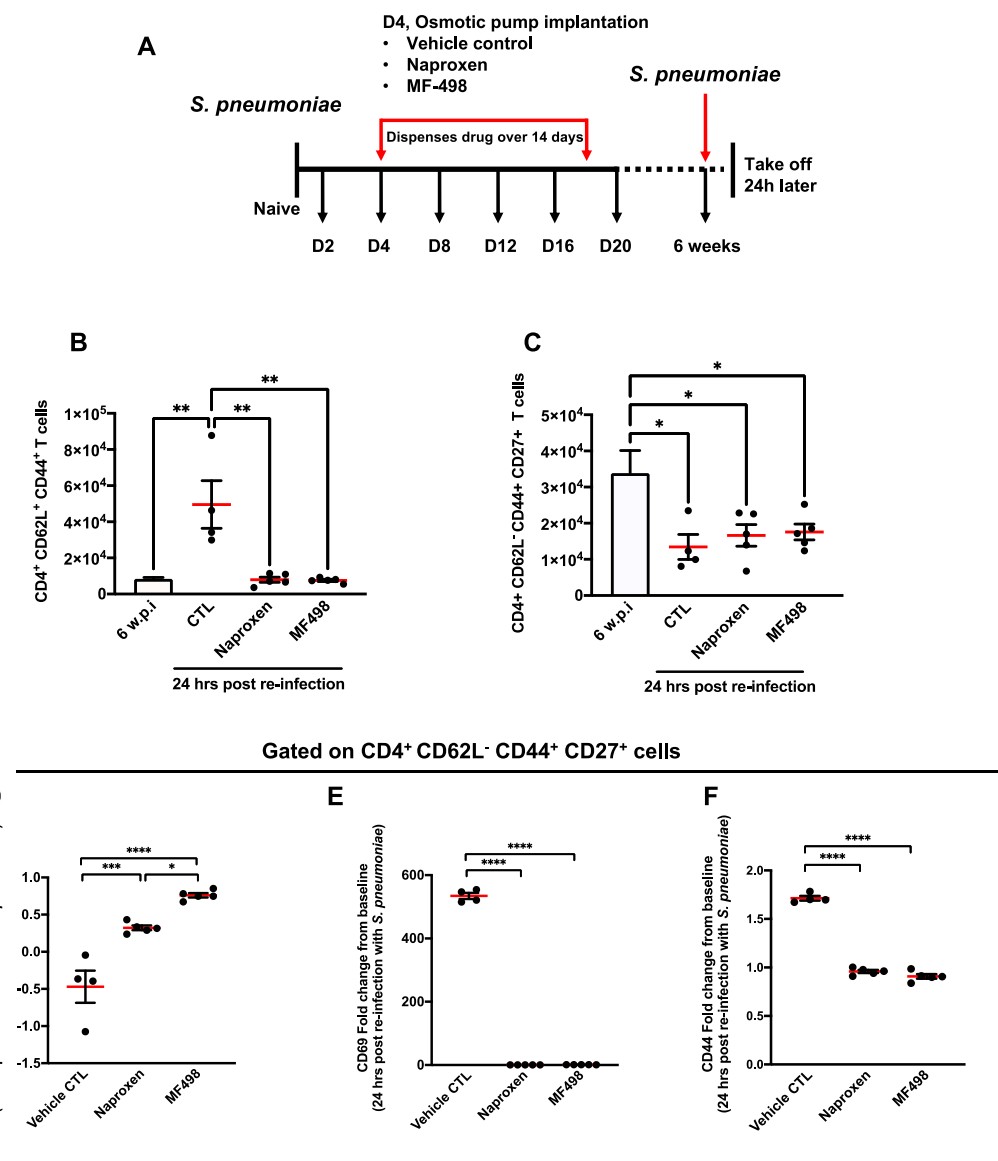

Gated on CD4$^+$ CD62L$^-$ CD44$^+$ CD27$^+$ cells

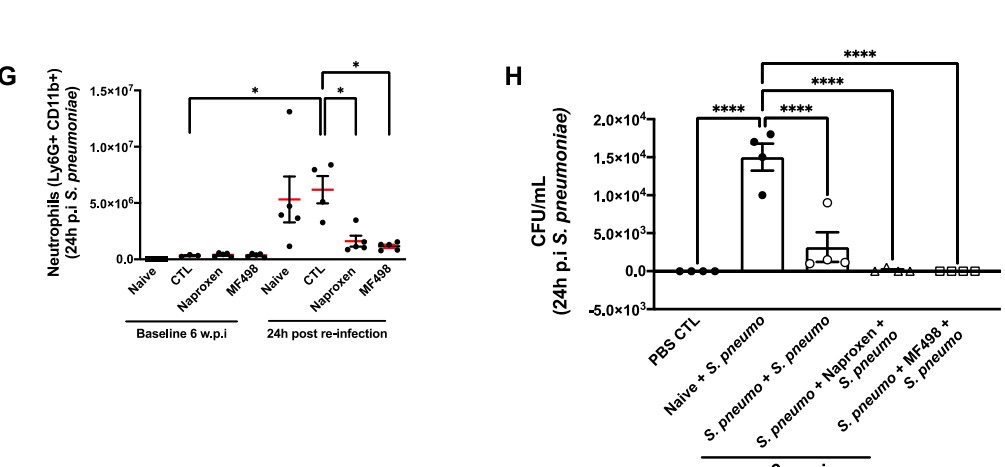

myofibroblast differentiation[19–25]. Therefore, the molecular basis for the increased lung fibrosis following PGE$_2$ inhibition may arise from a combination of the impact inflammatory macrophages have on tissue integrity as well as a direct effect blocking pro-fibrotic PGE$_2$.

EP4 is the most predominant prostaglandin receptor expressed during post-resolution modulating T cells and preventing tissue damage, with EP1-3 being negligible. The functional role of EP4 is evidenced following the actions of the EP4 antagonist, MF498 and the pan cyclooxygenase inhibitor naproxen. Collectively, these data suggest that PGE$_2$ exerts is post-resolution effects through EP4. In terms of downstream signalling, while cAMP is classically activated by EP4[26], there is also evidence of PI3K/AKT involvement in EP4 signalling[27]. As the post-resolution immune landscape is littered with many subtypes of mononuclear phagocytes and T cells, discerning the precise

**Fig. 7 | Post-resolution PGE₂ via EP4 determine re-emergence of memory T cells in response to secondary infection. A** Minipumps loaded with either the pan COX inhibitor naproxen or the EP4 antagonist MF498 were implanted into mice post-resolution four days after intranasal *S. pneumoniae*. 6 weeks after the initial challenge (or 38 days after this intervention), these sensitised mice or their controls were challenged with *S. pneumoniae* and impact of this intervention on (**B**) CD4⁺/CD44⁺/CD62L⁺ and (**C**) CD4⁺/CD44⁺/CD62L⁻/CD27⁺ T cell numbers was determined 24 h later. Indeed, levels of expression of (**D**) CD103 and the T cell activation markers (**E**) CD69 and (**F**) CD44 was also determined on CD4⁺/CD44⁺/CD62L⁻/CD27⁺ T cells. Besides lymphocytes we determined how inhibition of PGE₂ synthesis or EP4 antagonism also impacted on the ability to recruit (**G**) neutrophils and ultimately clear the secondary challenge of (**H**) *S. pneumoniae*. Data were analysed by one-way analysis of variance (ANOVA) and Tukey's multiple comparisons test. A *p* value of <0.05 was taken as the threshold of significance with graphical representation as; $p < 0.05 = *$, $p < 0.01 = **$ and $p < 0.001 = ***$ and presented as mean ± SEM ($n = 4$–5 mice/group).

downstream signalling pathway transduced by EP4 is likely cell type specific and tissue niche specific and therefore beyond the scope of this report.

Taken together, these data raise concerns regarding the use of NSAIDs during infections. Indeed, there is evidence from case control studies that complications of respiratory infections may be more common when NSAIDs are used[28–35]. While such observational evidence is difficult to interpret due to confounding factors, where this has been controlled for, the association still persists[32,36,37]. In contrast, paracetamol, a non-COX inhibitor, is less likely to result in such complications[36]. The molecular mechanism behind these effects may range from a skewed Th1 response early in the infectious in addition to our data showing that post-resolution prostaglandins limit tissue injury and lays down TrM and TcM. This means that in addition to provoking tissue injury, the widespread use of NSAIDs may be unwittingly compromising the immune system's role of bolstering tissue immunity following infection.

Central to this research is that pro-resolution processes lead to a sequence of events that ultimately benefit the host by shaping tissue immunity and limiting tissue damage. By corollary, blocking resolution will not only cause overt changes such as tissue injury, but as shown in this study, it will also leave the tissue bereft of protective lymphocytes, a more conspicuous but equally undesirable effect. While it is important to understand the factors that drive resolution and resolution's subsequent immune events, it is equally important to understand how pathogenic infections dysregulate these pro-resolution pathways and drive chronic inflammation. Hence, there is hitherto unappreciated immune activity following inflammation that has a long-term impact on tissue immunity and host defence. And while uncovering the molecular basis of these events will add a new chapter to inflammation biology, there are long-term consequences of non-resolved infections that profoundly remodel the immune system thereby contributing to the rising incidence of autoimmune and inflammatory disorders affecting barrier tissues, such as inflammatory bowel disease, asthma, allergy and psoriasis[38,39].

## Methods

### Mice

All experiments involving animals were approved by the university College London Animal Welfare and Ethical Review Board (AWERB) and carried out in accordance with UK Home Office regulations (Project licence P69E3D849). Wild-type 8–10-week-old male C57BL6/J mice were obtained from Charles River (strain code 632) and and housed under specific pathogen-free conditions in individual ventilated cages (IVCs) at a temperature of 21 °C with water and food *ab libitum*. All procedures were carried out under the UK's Home Office Animals (Scientific Procedures) Act 1986. All in vivo work was carried out using live *Streptococcus pneumoniae* strain EF3030 (19F). Bacterial stocks of *S. pneumoniae* were grown and supplied by Jeremy Brown (UCL Respiratory). Briefly, bacterial colonies were isolated from individual CFUs grown on blood agar (Tryptic Soy Agar (Becton Dickinson) plates and incubated in Tryptic Soy Broth (TSB; Becton Dickson) at 37 °C with caps unscrewed (if incubator is 5% CO2) until the optical density (OD) was 0.3–0.5. Concentration of CFU was determined using OD values and a standard curve. Bacteria were supplemented with 15% glycerol and stored at −80 °C until use.

Mice were anaesthetised with isoflurane and acute lung inflammation was induced by intranasal (i.n.) infection with 40 μL of *S. pneumoniae* ($2 \times 10^7$ CFU) in sterile phosphate buffered saline (PBS; Corning). For in vivo re-infection studies mice were re-infected with *S. pneumoniae* ($2 \times 10^7$ CFU) 14 or 42 days post initial infection. For ex vivo re-infection studies of whole lung cells or isolated T cells, heat-killed *S. pneumoniae* (heated at 65 °C for 30 min; $10^5$ CFU) was used.

Oral dosing (o.d.) and intraperitoneal (i.p.) injections were performed without anaesthesia using methylcellulose (0.5% in sterile water; Acros Organics) and PBS, respectively, as a vehicle.

The ALZET osmotic micro-osmotic pump (model 1002) delivers solutions continuously for 14 days without the need for frequent dosing and handling of animals. For immediate pumping upon implantation, the prefilled pumps were incubated in 0.9% saline at 37 °C for 4–6 h. Pumps were implanted subcutaneously via a small incision made in the skin between the scapulae. For re-infection studies mice were re-infected with *S. pneumoniae* ($2 \times 10^7$ CFU) 14 or 42 days post initial infection.

Mice were culled at experimental endpoints via i.p. injection of pentobarbital (1 mg/kg) or inhalation of increasing concentrations of CO₂. Secondary confirmation of death was performed via exsanguination.

### Pharmacological agents

For monocyte depletion studies, intraperitoneal (i.p.) injections of 15–20 μg of the anti-CCR2 monoclonal antibody, MC-21 (supplied by Matthias Mack, Department of Internal Medicine, University Hospital Regensburg, Germany), were performed every 48 h over a maximum period of five days to prevent the generation of neutralising antibodies to MC-21.

To investigate the effect of prostanoid blockade, naproxen (Sigma), a COX inhibitor, was dosed either orally or via osmotic pump implantation at a dose of 20 mg/kg daily. The EP4 receptor antagonist MF498 (30 mg/kg, Cayman Chemical) was delivered via osmotic pump infusion over a period of 14 days.

### Tissue processing, flow cytometry and cell sorting

Isolated lungs were digested in 150 U/mL collagenase type IV (Sigma) supplemented with DNase (1 mg/mL), at 37 °C for 45 min before being passed through a Falcon 70 μM cell strainer. Red blood cells were lysed (ACK lysis buffer) and whole lung cells were blocked with anti-mouse CD16/32 (Tru-stain FcX™; Biolegend) and stained with fluorochrome-conjugated antibodies against surface markers at a concentration of 1:100 unless otherwise stated: CD3-PE (17A2), CD4-APC (GK1.5), CD8a-V500 (53.6.7), CD11b-FITC/PerCP-Cy5.5 (M1/70), CD11c-BV605 (N418), CD19-FITC/PE (6D5/1D3), CD24-PE-Cy5 (M1/69), CD25-BUV395 (PC61), CD27-PE-Dazzle (LG.3A10), CD44-PE-Cy7 (IM7), CD45-BV711/V500 (30-F11), CD49a-PerCP-Cy5.5 (HMa1), CD64-BV421/APC (X54-5/7.1), CD62L-BV605 (MEL-14), CD69-BV421 (H1.2F3), CD103-AF700 (2E7), F4/80-PE (BM8), Ly6C-BV510/V450 (HK1.4), Ly6G-BUV395/PerCP-Cy5.5 (1A8), LYVE-1-PE-Cy7 (ALY7), MerTK-FITC (2B10C42), MHC-II-AF700 (1:75, M5/114.15.2), NK1.1-BV786 (1:75, PK136) and SiglecF-APC/PE-CF594 (517007 L).

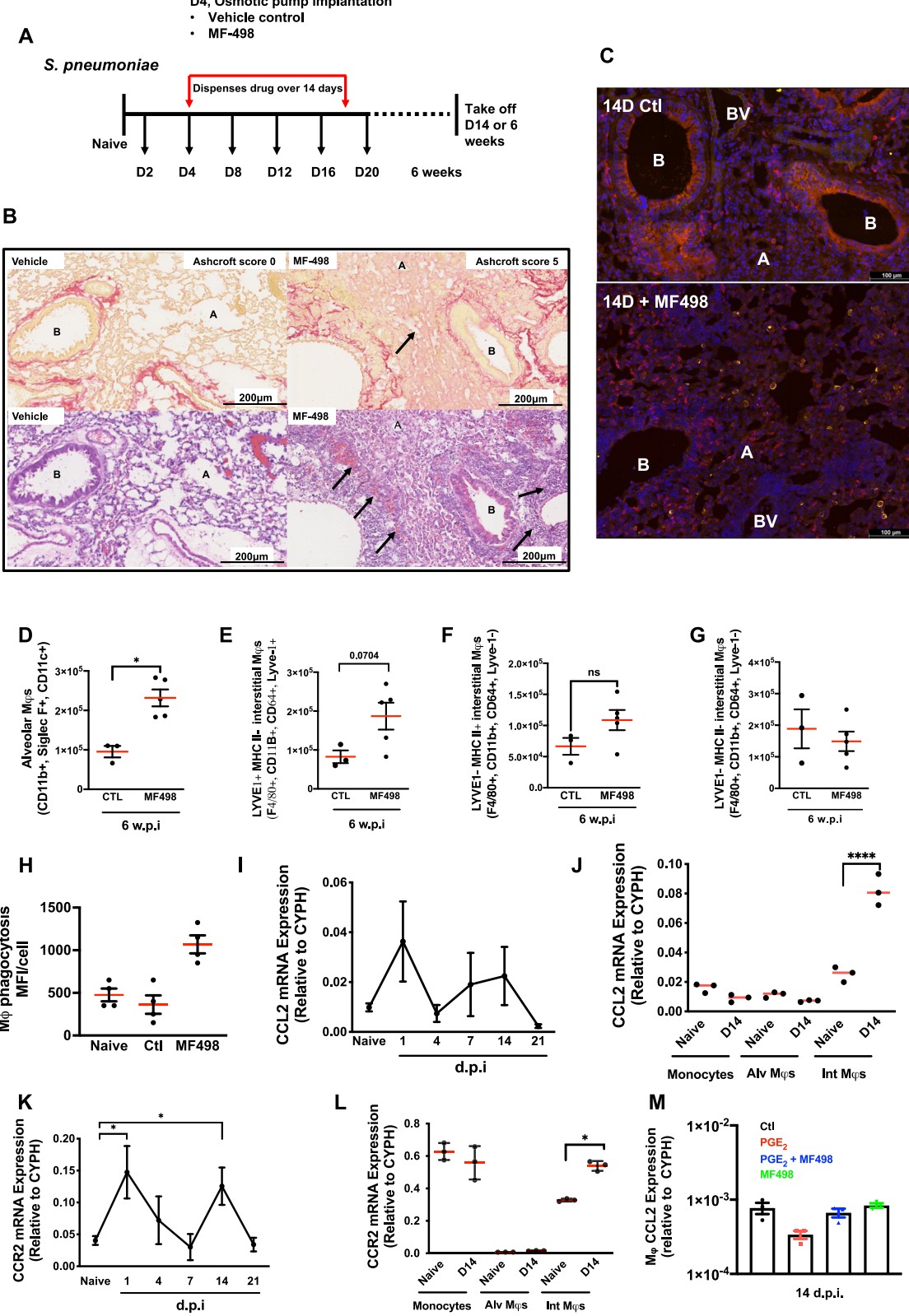

**Fig. 8 | Post-resolution PGE₂ via EP4 prevents excessive tissue injury and regulates macrophage trafficking.** The dosing regime in (**A**) revealed the impact of therapeutically antagonising post-resolution EP4 on (**B**) fibrosis along with examples of vascular occlusion (arrows) and (**C**) macrophage infiltration by confocal microscopy as well as (**D–G**) flow cytometry at day 14. These infiltrated macrophages were examined for their (**H**) phagocytic ability. The infiltration of post-resolution macrophages was associated with a second wave of (**I**, **J**) CCL2 expression with its receptor (**K**, **L**) CCR2 being negatively controlled by (**M**) EP4. A Alveoli, B bronchiole, BV blood vessel. Students unpaired *t*-test was used to compare the means of two groups. Differences between multiple groups were analysed using one-way analysis of variance (ANOVA) and Tukey's multiple comparisons test. A *p* value of <0.05 was taken as the threshold of significance with graphical representation as; $p < 0.05 = *$, $p < 0.01 = **$ and $p < 0.001 = ***$ and presented as mean ± SEM ($n = 3–5$ mice/group).

For intracellular cytokine staining, lung cells ($10^6$ cells per well) were seeded in 96-well round bottom plates and stimulated or not with PMA (50 ng/ml)/ionomycin (500 ng/ml) or $10^5$ CFU of heat-killed *S. pneumoniae* (heated at 65 °C for 30 min) for 4–24 h. All samples were treated with a protein transport inhibitor containing brefeldin A (Biolegend, 1x) during the final 4 h of incubation. After stimulation, cells were stained with a viability marker (Zombie UV™, Biolegend), blocked with anti-mouse CD16/32 (Tru-stain FcX™; Biolegend) and stained with fluorochrome-conjugated antibodies against surface markers at a concentration of 1:100: CD45-BV785 (30-F11), CD3-APC-Cy7 (17A2), CD4-APC (GK1.5), CD8a-V500 (53.6.7), CD27-PE-Dazzle (LG.3A10), CD44-PE-Cy7 (IM7), CD49a-PerCP-Cy5.5 (HMa1), CD62L-BV605 (MEL-14), CD69-BV421 (H1.2F3), CD103-AF700 (2E7). Cells were fixed with PFA 4% for 15 min before permeabilisation with a permeability buffer (Biolegend). Cells were then incubated with intracellular antibodies against IL-17 (FITC, TC11-18H10.1), TNF-α (BV711, MP6-XT22) and IFN-γ (PE, XMG1.2), all obtained from Biolegend. Unstained samples and Fluorescence-minus-one (FMO) were used as a control. In some experiments, CountBright™ Absolute Counting Beads (Thermo Fisher Scientific) were added to the samples before data were acquired on a BD Fortessa UV X20 flow cytometer using BD FACSDIVA software. Compensation and data analysis were performed using FlowJo software (v10.7.2). Cell sorting was performed on FACSAria II (BD Biosciences).

## Immunohistochemistry of lung sections

Mouse lungs were dissected from the thoracic cavity and placed in 3 mL Hank's Balanced Salt Solution (HBSS; 1X, No calcium or magnesium, Gibco) on ice. Fatty tissue was dissected away and the whole left lobe was fixed by placing it in PBS 4% paraformaldehyde (PFA) at RT for 1–4 h. Following fixation, tissue was briefly immersed in PBS to wash off extra fixative then immersed in 70% ethanol. Tissues were dehydrated and infiltrated with paraffin wax using an automated tissue processor (Leica). Tissues were then embedded in paraffin wax.

Prior to staining, slides were incubated at 65 °C for 30 min to melt excess paraffin wax. Slides were then further dewaxed in an autostainer (Leica). Following dewax, sections were briefly immersed in distilled water and incubated in PBS for 10 min at RT. Antigen retrieval was achieved by breaking the formalin cross links with a sodium citrate solution (0.3 % sodium citrate (Sigma Aldrich), 0.05 % TWEEN 20 (Sigma Aldrich), pH 6) at 95 °C in the water bath for 13 min. Following antigen retrieval, sections were washed with PBS. Next, the sections were permeabilised and blocked for endogenous peroxidase activity by incubation with 3% hydrogen peroxide (VWR Chemicals) in methanol (VWR Chemicals) for 10 min at room temperature. The sections were then washed twice with PBS. Haematoxylin and eosin (H&E) staining was carried out as per the manufacturer's instructions (Abcam, ab245880). For collagen staining slides were washed in PBS and stained with Pico-Sirius red solution (Abcam) and incubated for 60 min at RT. Slides were rinsed quickly in two changes of acetic acid solution (0.5%) and dehydrated in two changes of absolute alcohol before mounting in synthetic resin (xylene). For GR1 staining, sections were blocked for 1 h at RT followed by primary antibody staining (anti-GR1) overnight. Sections were stained with a HRP-conjugated secondary antibody and GR1 was visualised using a Diaminobenzidine chromogen solution. Isotype control antibody was used to control for non-specific staining. Images were captured with a digital slide scanner (Nanozoomer) and analysed using Nanozoomer digital pathology software.

Scoring of immunohistochemically stained murine lung sections was conducted using the Ashcroft scale[40]. Briefly, each section was assigned a score between 0 and 8, indicating no fibrosis or maximal severity, based on prevalence of histological features.

## Immunofluorescence of lung sections

Mouse lungs were dissected from the thoracic cavity and placed in 3 mL Hank's Balanced Salt Solution (HBSS; 1X, no calcium or magnesium, Gibco) on ice. Fatty tissue was dissected away and the whole left lobe was fixed by placing it in PBS 4% PFA at RT for 1–4 h. Samples were dehydrated prior to embedding in OCT. Sections (8–10 μM) from embedded lungs were rehydrated in PBS for 10 min and incubated in blocking buffer (1% BSA, 1% FCS, 0.3% TX100 in PBS) for 1 h at RT. Primary antibodies (Alexa Fluor® 647 anti-mouse F4/80 [BM8, Biolegend], Alexa Fluor® 488 anti-mouse CD3 [17A2, Biolegend]) were diluted in blocking buffer and applied to sample for 1 h at RT. Samples were wash 3 times in PBS, then counter stained with DAPI (300 nM in dH2O) for 5 min at room-temperature. Slides were mounted with SlowFade mount and sealed. Slides were stored at 4 °C, protected from light. Images were generated on a Leica SP8 confocal microscope, with a DM6000 microscope, at 40X. Images were captured using LAS X software.

To analyse lung lipid biosynthesis machinery and EP receptor expression, OCT embedded tissue sections (8–10 μM) were sequentially immersed in pure acetone and pure ethanol for 10 min each, followed by incubating in blocking buffer (1% BSA, 1%FCS, 0.3% TX100 in PBS) for 1 h at RT in a humidity chamber. For day 14 samples, sections were incubated with primary conjugated antibodies, Alexa Fluor® 488 anti-mouse F4/80 [BM8, Biolegend], Alexa Flour® 467 anti-mouse COX-2 [EPR3777, Abcam-Conjugated using Thermo Fisher labelling kit A20186], CF®555 anti-mouse CD3 [SP162, Abcam-Conjugated using Biotium Mix-n-Stain™ CF® Dye Antibody Labelling Kits] and primary unconjugated antibodies anti-mouse EP4/PTGER4 (4A2A12, Proteintech) anti-human mPGES-1 (Polyclonal, Cayman Chemical 160140) and anti-mouse Siglec F/CD170 (S17007L, Biolgend) overnight at 4 °C. Samples were washed 3 × 5 min in PBS and primary unconjugated antibodies were detected using antigen specific IgG used at 1:200: Alexa Fluor® 568 anti-goat IgG, Alexa Fluor® 647 anti-rat IgG, Alexa Fluor® 488 anti-mouse IgG. Samples were counterstained with DAPI (300 nM in dH2O) for 5 min at RT. Slides were mounted with SlowFade mount and sealed. Images were generated on a Leica SP8 confocal microscope, with a DM6000 microscope, at 20X or 40X. Images were captured using LAS X software.

For naïve, 24 h and 48 h timepoints, sections were stained in 4 rounds. Staining rounds used the following antibodies.

1. Alexa Fluor® 488 rat anti-mouse CD3 [17A2, BioLegend], Alexa Fluor® 647 rat anti-mouse F4/80 [BM8, BioLegend], Unconjugated rabbit anti-human mPGES-1 [polyclonal, Cayman Chemicals 160140].

2. Rabbit anti-mouse CCL2 [polyclonal, PTG Lab 25542-1-AP–conjugated using Biotium Mix-n-Stain™ CF-488 Dye Antibody Labelling Kits], Rat anti-mouse Siglec-F [S17007L, BioLegend – conjugated using Biotium Mix-n-Stain™ CF-555 Dye Antibody Labelling Kits], rabbit anti-mouse COX-2 [EPR3776, Abcam-conjugated using Invitrogen Alexa Fluor® 647 Antibody Labelling Kits].

3. Alexa Fluor® 488 rat anti-mouse LYVE-1 [ALY7, Thermo Fisher], mouse anti-mouse EP4/PTGER4 [4A2A12, Proteintech–conjugated using Biotium Mix-n-Stain™ CF-555 Dye Antibody Labelling Kits], Alexa Fluor® 647 rat anti-mouse CD326/EpCAM [G8.8, BioLegend].

4. Alexa Fluor® 488 rat anti-mouse Vimentin [W16220A, BioLegend].

Primary stains were performed overnight at 4 °C. Samples were washed 3 × 5 min in PBS. For the first round, this was followed by 1:200 goat anti-rabbit Alexa Fluor® 555 for 1 h at room temperature. This was washed 3 × 5 min in PBS before counterstaining with DAPI (300 nM) for 5 min at RT. Slides were mounted with 90% glycerol, 4% propyl gallate in PBS and coverslipped. Images were generated on the GE Healthcare Cell Dive™ IN Cell 2500 platform at 20X. Images were analysed using QuPath (v0.4.4). In between rounds, coverslips were removed by immersing slides in PBS before signal was bleached using 30% $H_2O_2$ in 0.1 M $NaHCO_3$ (stock 0.5 M at pH 10.9–11.3) for 15 min, twice. Before

each image acquisition run, a background image stained only using DAPI was acquired. Background correction, alignment and image restitching was done in Cell Dive™ software.

All images from across all time points were generated under identical conditions, including same antibody/antibody concentration and microscopy settings to ensure that images from all time points are comparable.

## Quantitative polymerase chain reaction (qPCR)

mRNA was extracted from whole lung tissue or cell sorted populations and contaminating DNA was removed by DNase I (Thermo Fisher). Reverse transcription (RT)−PCR was used to generate complementary DNA (cDNA). Specific genes (Primer sequences shown in Supplementary Table 1) were amplified and quantified by PCR using by Brilliant III Ultra-fast SYBR Green (Agilent technologies) or power SYBR Green (Thermo Fisher Scientific) on an AriaMX Real-Time PCR system (Agilent) under the following conditions: 95 °C for 30 s followed by 40 cycles of 55 °C for 5 s and 60 °C for 30 s. A master mix was prepared for a single reaction. All samples were run in duplicates and the relative amount of mRNA was calculated using the comparative Ct method and normalised to the expression of cyclophylin.

## Bulk transcriptomics of lung macrophages

RNA sequencing was performed by UCL Genomics (London, UK). Total RNA was extracted, and cDNA libraries were prepared using reagents and protocols supplied with the NEBNext Low Input mRNA kit. Libraries were sequenced using a NovaSeq SP v1.5 (Illumina) (100 cycles, -18 million reads/sample, 1.8GB/sample).

Quality control of the FASTQ.files was performed on FASTQC Version 0.11.9 (https://www.bioinformatics.babraham.ac.uk/projects/fastqc/). Basic Local Alignment Search Tool (BLAST) (https://blast.ncbi.nlm.nih.gov/Blast.cgi) was used to identify contaminations, including primers. FASTQ files were aligned to the mouse genome (GRCm38) using the alignment programme HISAT2 version 2.1.0 resulting in subsequent BAM files. SAMTOOL version 1.9 was then used to convert BAM files to sorted SAM files. Through subread version 2.01, featureCounts was used to map the read locations to the genomic regions using the GRCm38 construct. RStudio version 1.4.1106, DESeq2 version 1.32.0 and edgeR version 4.0.5 were used to identify differential gene expression from the mapped files. R packages ggplot2 version 3.3.3 and enhanced volcano version 1.10.0 were used to visualise the data. Differentially expressed genes were used with Protein Analysis Through Evolutionary Relationships (PANTHER) Classification System version 16.0 (http://www.pantherdb.org/about.jsp) to identify enriched GO-Terms with statistical significance. Similarly, Reactome version 76 (https://reactome.org/what-is-reactome) was used to identify enriched pathways with statistical significance.

## Lipidomics

Prostanoids and hydroxy fatty acids were extracted from whole lung tissue as described[41,42]. Briefly, lung tissue was homogenised in methanol, then diluted in water to a final concentration of 15 % (v/v) methanol/water. Internal standards were added (20 ng each of $PGB_2$-$d4$, 12-HETE-$d8$, 8,9DHET-$d11$, 8(9)EET-$d11$ and 12(13) DiHOME-$d4$; Cayman Chemical, Ann Arbor, USA). Samples were semi-purified using solid-phase extraction (500 mg C18-E cartridges; Phenomenex, Macclesfield, UK). Analytes were eluted in methyl formate, dried under nitrogen and reconstituted in ethanol, then stored at −20 °C until analysis. Analyte separation was performed using ultraperformance liquid chromatography (Acquity; Waters, Wilmslow, UK) and a C18 column (Acquity UPLC BEH; 1.7 μm; 2.1 × 50 mm; Waters, Wilmslow, UK), before multiple reaction monitoring on a triple quadrupole mass spectrometer with electrospray ionisation (Xevo TQ-S; Waters, Wilmslow, UK).

Quantitation was performed using calibration lines constructed with commercially available standards (Cayman Chemicals, Ann Arbor, UK).

## Ex vivo functional T cell Assays

Spleens from WT C57BL/6 naïve mice were passed through 40 μm cell strainers, followed by lysis of red blood cells with Gibco™ ACK Lysing Buffer. Cells were washed, pelleted and suspended in RPMI containing FBS 10%, L-Glutamine and antibiotics. Splenocytes were plated ($2 \times 10^6$ cells per well) on 12-well plates previously coated with ultra-LEAF Purified anti-mouse CD3ε Antibody (3 μg/mL; Clone: 145-2C11; Biolegend). Ultra-LEAF Purified anti-mouse CD28 Antibody (3 μg/mL; Clone: 37.51; Biolegend) was added to all the wells, and cells were incubated for 24 h at 37 °C; 5% $CO_2$. Cells were then pre-treated with the EP4 receptor antagonist MF498 (10 μM; Cayman Chemical) or vehicle for 1 h before the treatment with 15(S)−15methyl $PGE_2$ (50 ng/ml; Cayman Chemical) and/or rmTGFb1 (10 ng/mL; R&D systems) or vehicle. DMSO in RPMI was used as the vehicle for both MF498 pre-treatment and $PGE_2$ and/or TGF-β treatment. On day 5, cells were treated with a protein transport inhibitor containing brefeldin A (Biolegend) for 4 h. Cells were collected and stained for extracellular and intracellular flow cytometry as described above.

## Statistics and reproducibility

GraphPad Prism software (Version 8.2.1) was used for the generation of figures and statistical analyses. Data are presented as mean ± standard deviation (SD). The unpaired data were tested for Gaussian distribution using normality tests recommended by the software. Students unpaired t-test was used to compare the means of two groups. Differences between multiple groups were analysed using one-way analysis of variance (ANOVA) and Tukey's multiple comparisons test. A $p$ value of <0.05 was taken as the threshold of significance with graphical representation as; $p < 0.05 = *$, $p < 0.01 = **$ and $p < 0.001 = ***$.

Experiments were designed with advice from a statistician using power analysis to use the minimum number of mice per group to detect significance differences. We used historical data to inform on power calculations. Based on a paired t-test, our experiments will have >80% power (lowest 81.9%) to detect the expected mean difference between groups at the 5% significance level. The estimated magnitude of differences between groups and standard errors of the mean are based on previous experiments. To mitigate non-reproducible results, experiments were repeated with at least 2 independent experiments and key experiments with 3 independent repeats. Experiments were blinded where practical - Moreover in all cases blinded and unblinded, we routinely take several steps to avoid bias, including (1) mice are housed in cages of 4 mice per cage with experimental/control animals co-housed allowing for social interaction. (2) To avoid circadian rhythm variances, experiments are scheduled at the same time each day.

Mass spec and RNAseq data collection was not replicated but each group contained $n = 5$ replicates per group and samples were randomly allocated a number following sample collection. Data analysis was acquired blinded.

## Reporting summary

Further information on research design is available in the Nature Portfolio Reporting Summary linked to this article.

# Data availability

Data generated in this study are provided in the Supplementary Information/Source Data File. Remaining raw data can be found at the following respective links -Lipidomics: https://doi.org/10.48420/25285396. Flow cytometry data: https://doi.org/10.5281/zenodo.10611080. RNAseq: GSE264058. Source data are provided with this paper.

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

## Acknowledgements

The authors wish to acknowledge the financial support provided by a BBSRC-AstraZeneca/CASE studentship. We also wish to thank Neil O'Hara (University of Manchester) for excellent technical support. The authors also wish to acknowledge the expertise and technical guidance of the Digital Pathology Omics (DPOC), Nuffield Department of Orthopaedics, Rheumatology and Musculoskeletal Sciences. University of Oxford.

## Author contributions

K.T.F. independently carried out this work as part of her PhD thesis. H.E.B. analysed/interpreted transcriptomic data. C.M.M., J.S.W., J.F. and RPHDM characterised all antibodies for and carried out confocal and imaging experiments. M.M. supplied and advised on the use of MC-21. J.B. and G.E. established the lung challenge model. A.N.A. provided invaluable input on T-cell biology. M.A.S., J.R.W.G., P.J. and O.V.B. assisted with animal challenge models. A.N. and A.C.K. carried out lipidomic analysis. D.W.G. designed and oversaw the execution of the paper. All authors contributed to the drafting, reviewing and final approval of the manuscript.

## Competing interests

The authors declare no competing interests.

## Additional information

[1]Department for Experimental and Translational Medicine, Division of Medicine, 5 University Street, University College London, London WC1E 6JJ, UK. [2]Nuffield Department of Orthopaedics, Rheumatology and Musculoskeletal Sciences, Botnar Research Centre, Windmill Road, University of Oxford, OX3 7LD Oxford, UK. [3]Translational Science and Experimental Medicine, Research and Early Development, Respiratory and Immunology, BioPharmaceuticals R&D, AstraZeneca, Cambridge, UK. [4]Universitätsklinikum Regensburg, Innere Medizin II/Nephrologie-Transplantation, Regensburg, Germany. [5]UCL Respiratory, Division of Medicine, 5 University Street, University College London, London WC1E 6JJ, UK. [6]Laboratory for Lipidomics and Lipid Biology, Division of Pharmacy and Optometry, School of Health Sciences, The University of Manchester, Oxford Road, Manchester M13 9PT, UK. [7]Present address: School of Cancer and Pharmaceutical Sciences, King's College London, London SE1 1UL, UK. [8]Present address: Centre for Sports, Exercise and Life Science, Coventry University, Priory St, Coventry CV1 5FB, UK. ✉e-mail: d.gilroy@ucl.ac.uk

