## [Peer Review File · Nature Communications]

Post-resolution macrophages shape long-term tissue immunity and integrity in a mouse model of *Streptococcus pneumoniae*REVIEWER COMMENTS

Reviewer #1 (Remarks to the Author):

Title: "Post-resolution macrophage-derived lipids shapes long-term tissue immunity and integrity"
This work contributes to elucidating the role of lung macrophages, especially alveolar macrophages, in the resolution phase of the inflammatory response in mice caused by pulmonary infection with *S. pneumoniae*. The authors showed that during post-resolution phase, lungs exhibited increased numbers of alveolar macrophages that express COX-2 and mPGES-1 mRNA required for PGE2 biosynthesis and that PGE2 and its receptor EP4 play a primary role in infiltration of lungs with early effector T cells CD4⁺/CD44⁺/CD62L⁻/CD27⁺ (TeME) including TeME expressing lung homing integrin CD103. Importantly, they also showed that blocking of PGE2/EP4 pathway leads to macrophage infiltration and tissue fibrosis in the lungs. Thus, the authors presented a set of in vivo experiments that reveal a novel role for PGE2/EP4 signaling in resolution of infection in the lungs. The conclusions of the study are well supported by the provided data. For comments and minor corrections and, please see below

1. Figure 4 Treatment with anti-MC-21 Ab reduced macrophages in the lungs. The authors should determine whether PGE2 in the lungs was reduced after anti-MC21 treatment to confirm the source of PGE2.
2. Figure 5 The mechanism of PGE2 induction in lung macrophages should be investigated, whether PGE2/COX-2 was upregulated in response to bacteria or in response to IL-6, TNF α , IL-10, TGF- β cytokines that were shown to be increased during acute phase of the infection and post-resolution.
3. Figure 5 The presence of COX-2 and mPGES1 is shown for alveolar and interstitial macrophages on Day 14 only. However, Figure 2 showed that alveolar and interstitial macrophages were increased in the lungs at earlier time points and may be as early as D1. The authors should investigate whether COX-2 and mPGES1 were expressed in macrophages early after infection.
4. The discussion will benefit from shortening and being more concentrated around the subject of the study
5. Page 9, reference to Figure 3D in sentence "Mononuclear phagocytes sorted by FACS from naïve and post-resolution lungs revealed that post-resolution alveolar macrophages were enriched with COX-2" should be replaced with Figure 5D

Reviewer #2 (Remarks to the Author):

This is an interesting study that aims to investigate the mechanisms of resolving inflammation and immune-mediated tissue repair that occur after a *S. pneumoniae* pneumonia. The authors identify a new pathway mediated by PGE2/EP4, that induce an influx of antigen specific lymphocytes with a memory phenotype, as well as macrophages with either an alveolar or interstitial phenotype. Inhibition of this late surge in immune cell infiltration leads to tissue fibrosis, demonstrating the relevance of this process. The findings of this study are relevant and new, and the manuscript is clearly written.

Comments:

1. It is unclear whether all experiments were performed only one time, or did the authors repeat and validate the most important findings in independent experiments.
2. In Figs.2B-E there are 8 mice assessed on day 14, but in Fig.2F-H (presumably the same experiment?), there are only 5 mice analyzed on day 14. Why this discrepancy?
3. In Fig.1G, the second peak of monocyte infiltration in the lungs on day 14 is heterogeneous: in 5 mice the peak occurred, while in 3 mice it did not. Similar heterogeneity is seen in Figs. 2D and 2E. Can the authors hypothesize the cause and eventually the consequences of this heterogeneity?

4. Based on the transcriptome changes presented in Fig.3, the authors suggest that 'robust changes' are present in alveolar macrophages. As only 33 genes have shown changes, I suggest to be cautious and not overinterpret the data.

Reviewer #3 (Remarks to the Author):

This is a new report that describes the role of diverse monocyte/macrophages lineages on inflammation and their subsequent resolution on injury in the lung compartment. They characterize the macrophage populations and the chronicity of macrophage influx and subsequent resolution. They further attempt to elucidate the role the eicosanoid pathway may play in these processes. While the report is intriguing several critical questions remain.

Major Comments:

- 1) Of paramount concern is the use of live strep (?) for pilot studies and the transition to heat-killed strep for the therapeutic intervention studies (Fig. 6). It is not clear why the model was changed for select studies. Was heat killed strep used for all the studies or just the studies on Figure 6. This should be clarified and clearly described throughout the manuscript.
- 2) This manuscript acknowledges prior work by Peters-Golden that stated that PGE2 attenuated macrophage efferocytosis after infection. PGE2 is also reported to reverse myofibroblast differentiation which contributes to the progression of the fibrotic response. Consequently, how do the authors explain that the pan inhibition of PGE2 with naproxen paradoxically worsens injury.
- 3) The data presented in Figure 5 does not show significant changes in PGE2. The authors do show changes in the RNA levels of proteins, which regulate PGE2 synthesis. These studies are limited to expression in macs. The authors over rely on RNA analyses as a surrogate for direct determination of PGE2 source and expression. This reliance continues throughout the manuscript for multiple targets. Further, as resident epithelial cells and fibroblasts likely contribute to the total PGE2 concentration, what is the relevance of the expression of COX-2, pGES1 etc by select cell populations in the contribution to the overall PGE2 production?
- 4) There is similar concern for the expression of the relevant EP receptors, such as EP4, on the surface of the different subsets of macrophages. Once again, the authors over rely on RNA assays as a surrogate for direct expression of EP4 or other EP receptors on the cell surface. Further, the data presented shows increases in EP4 mRNA but these changes are only significant for interstitial macrophages. The only way this data has any import is if the EP4 protein expression is also increased. Further, the authors would need to confirm the converse, activation of this select receptor can promote their selected processed.
- 5) While the expression of CCL2 may have been increased in the macrophage populations, epithelial cells are a likely contributor to CCL2. However, all of the assays focus on these cells exclusively. It is likely that local cellular changes in CCL2 expression only modestly impact more global changes in CCL2 in the lung compartment. Total CCL2 assays should be performed.

Minor comments.

- 1) More specific details are required for the *S. pneumoniae* dosing studies. For example, was the Strep administered from frozen aliquots at the established dose or was the strep actively in log phase growth when administered to the mice. This is a subtle detail but maybe critical in observations.
- 2) In Figure 8B, Ashcroft scores were provided in the figures; however, there is no apparent quantitation of these scores for differences in the sample groups. Further I do not believe that the vehicle control adequately represents an Ashcroft Score of 0. An uninjured control should also be provided to show a true "0".
- 3) It is not clear what the data in Supplementary Figure 4a is providing. Are these quantified values of the prostaglandins and family members via ELISA (Concentrations), qPCR (relative quantities) etc?
- 4) This manuscript would be greatly improved if histological sections were provided for the MC-21 treatment studies as well to show a physiological effect of monocyte/macrophage depletion.
- 5) MF498 does not antagonize PGE2. It is a specific EP4 blocker. Studies should be performed using EP2 and EP4 selective agonist and antagonist in splenocytes to confirm effect.

6) While both EP2 and EP4 activate cAMP, EP4 can differentially activate cAMP and PI3K/Akt signaling. Further, it has been reported that activation of the alternative EP4 signaling pathways could outpace the effects of the cAMP induction. Studies should be done to elucidate which aspect of EP4 mediates these effects. These studies could be limited to splenocytes.

REVIEWER 1

We thank this reviewer for their supportive comments.

1. Figure 4 Treatment with anti-MC-21 Ab reduced macrophages in the lungs. The authors should determine whether PGE₂ in the lungs was reduced after anti-MC21 treatment to confirm the source of PGE₂.

Author response: These samples have been analysed and found to result in a ~50% reduction in PGE₂ levels. These data have been included in **Supplementary Figure 6** and described on page 10 of the revised manuscript.

2. Figure 5 The mechanism of PGE₂ induction in lung macrophages should be investigated, whether PGE₂/COX-2 was upregulated in response to bacteria or in response to IL-6, TNF α , IL-10, TGF- β cytokines that were shown to be increased during acute phase of the infection and post-resolution.

Author response: This is something that we are actively pursuing. We are mining for signals/soluble mediators that are upregulated just after resolution and before the second rise in prostanoid biosynthesis which is around day 4 post inoculation (plus/minus one day). Unlike our work in the mouse peritoneal cavity with zymosan where we found a role for IFN/IP-10 in monocyte recruitment (PMID: 28954232), we found very little in the way of known and obvious mediators that might prepare for the post-resolution phase in the lung following *S. pneumoniae*. Lipidomic analysis has revealed signals that are transiently elevated at this intermediate time point including 14, 15 EET and 12, 13 EPOME, which may play a role in signalling post-resolution. However, this is turning out to be a separate project.

3. Figure 5 The presence of COX-2 and mPGES1 is shown for alveolar and interstitial macrophages on Day 14 only. However, Figure 2 showed that alveolar and interstitial macrophages were increased in the lungs at earlier time points and may be as early as D1. The authors should investigate whether COX-2 and mPGES1 were expressed in macrophages early after infection.

Author response: Expression of COX 2 and mPGES-1 protein level and localisation is shown in **Supplementary Figure 5**. Reference to these data is include on page 10 of the revised manuscript.

4. Shortened discussion.

Author response: Agreed! The discussion has been shortened resulting in a more focused analysis of the data.

5. Page 9, reference to **Figure 3D** in sentence "Mononuclear phagocytes sorted by FACS from naïve and post-resolution lungs revealed that post-resolution alveolar macrophages were enriched with COX-2" should be replaced with **Figure 5D**

Author response: Apologies, this is now corrected.

REVIEWER 2

We also thank this reviewer for their positive comments.

1. It is unclear whether all experiments were performed only one time, or did the authors repeat and validate the most important findings in independent experiments.

Author response: Experiments were designed with advice from a statistician using power analysis of historical data to minimise numbers of mice per group to detect significance differences. Group comparisons were made with ANOVA, and between specific groups by unpaired t-test. Skewed data would be logarithmically transformed if necessary. To mitigate non-reproducible results, experiments were repeated with at least 2 independent experiments and key experiments with 3 independent repeats. LC-MS/MS lipidomic data and RNAseq data were not repeated, but each group contained n=5 replicates per group. Samples for these experiments were randomly allocated a number following collection and data analysis was done blinded with experimental groupings/condition re-ascribed after analysis. Experiments on rodents were blinded where practical. To avoid circadian rhythm variances, experiments are scheduled at the same time each day.

This experimental approach can be included in the methods section if necessary.

2. In Figs. 2B-E there are 8 mice assessed on day 14, but in Fig.2F-H (presumably the same experiment?), there are only 5 mice analyzed on day 14. Why this discrepancy?

Author response: In Figure 2B-E experiments were initially done on 6-8 mice per group looking at alveolar and interstitial macrophages. Given the re-appearance of interstitial macrophage at D14, separate experiments were then repeated with n = 5 mice per group to tease apart the three sub-populations of interstitial macrophage during inflammatory onset, resolution, and post-resolution.

3. In Fig.1G, the second peak of monocyte infiltration in the lungs on day 14 is heterogeneous: in 5 mice the peak occurred, while in 3 mice it did not. Similar heterogeneity is seen in Figs. 2D and 2E. Can the authors hypothesize the cause and eventually the consequences of this heterogeneity?

Author response: This reflects the limitations of the intranasal instillation model including variability in antigen instillation, size of lung affected versus un-affected tissue etc as well as the impact of longer timepoints (i.e. acute timepoints are tighter with more variability as we move further away from initial infection). An alternative, intratracheal instillation, was considered during experimental design. However, a major concern of intratracheal instillation, which involves a bolus delivery straight into the lung in a short period of time, is that this mechanical insult may overwhelm the normal immune response to infection and therefore produce results reflective of a non-*S. pneumoniae* induced inflammatory response. Intranasal instillation, therefore, despite its limitations, is more representative of natural infection of the lung tissue leading to an acute inflammatory response.

4. Based on the transcriptome changes presented in Fig.3, the authors suggest that 'robust changes' are present in alveolar macrophages. As only 33 genes have shown changes, I suggest being cautious and not overinterpret the data.

Author response: Agreed! We have changed "robust" to "distinct" on page 8.

REVIEWER 3

We also thank this reviewer for their positive remarks.

Major comments

1. Of paramount concern is the use of live strep (?) for pilot studies and the transition to heat-killed strep for the therapeutic intervention studies (Fig. 6). It is not clear why the model was changed for select studies. Was heat killed strep used for all the studies or just the studies on Figure 6. This should be clarified and clearly described throughout the manuscript.

Author response: Every intranasal infection experiment conducted in mice used live *S. pneumoniae* including therapeutic intervention studies (i.e., osmotic pumps, MF-498, naproxen), reinfection experiments etc. The only time heat killed *S. pneumoniae* was used was when cells were stimulated *in vitro*. The change to heat-killed for *in vitro* experiments was guided by failure of live *S. pneumoniae* to elicit a robust response in culture due to its direct toxic effect on cells. This is clarified on page 14 of Material & Methods under heading of “Mice” where we used live bacteria and on page 15 under heading of “Tissue processing, flow cytometry and cell sorting” for the *in vitro* experiments.

2. This manuscript acknowledges prior work by Peters-Golden that stated that PGE₂ attenuated macrophage efferocytosis after infection. PGE₂ is also reported to reverse myofibroblast differentiation which contributes to the progression of the fibrotic response. Consequently, how do the authors explain that the pan inhibition of PGE₂ with naproxen paradoxically worsens injury.

Author response: Indeed, it is correct that PGE₂ plays a beneficial role in the setting of fibrotic lung disease. This arises from the ability of PGE₂ to limit many of the pathological features of lung fibroblasts and myofibroblasts, including the ability to limit fibroblast proliferation, migration, collagen secretion and to limit TGFβ-induced myofibroblast differentiation. Hence, it would be expected that the inhibition of PGE₂ synthesis or antagonism of its receptor/s would worsen lung fibrosis as reported in this paper. What’s quite remarkable and novel here, is the extent of lung fibrosis relative to the comparatively mild, earlier transient inflammation and that it occurred after inflammation i.e. during the post-resolution phase.

3. The data presented in Figure 5 does not show significant changes in PGE₂. The authors do show changes in the RNA levels of proteins, which regulate PGE₂ synthesis. These studies are limited to expression in macs. The authors over rely on RNA analyses as a surrogate for direct determination of PGE₂ source and expression. This reliance continues throughout the manuscript for multiple targets. Further, as resident epithelial cells and fibroblasts likely contribute to the total PGE₂ concentration, what is the relevance of the expression of COX-2, pGES1 etc by select cell populations in the contribution to the overall PGE₂ production?

Author response: Levels of PGE₂ and PGI₂ (measured as 6-kets PGF1α) in **Figure 5A and H**, respectively, are replotted presenting individual data points, with the remainder of the extensive lipidomic profiles placed in **Supplementary Figure 4A-C**. In **Figure 5A and H** there is a trend towards a biphasic profile of these lipids over time, with PGE₂ dipping transiently at

day 4 post inoculation except for one data point. We suspect that this arises from the fact while whole lung was used to extract lipids for analysis by LC-MS/MS, only a proportion of the lung was affected by inflammation. To focus on the affected part of the lung we repeated experiments where we injected Evan’s blue prior to killing the mice – Evan’s blue binds to albumin, which moves into inflamed tissue highlighting inflamed tissue as blue. We dissected the blue/inflamed tissue away from the non-involved lung and extracted lipids for the measurement of PGE₂ 6-kets PGF1α, data here. This, more focused approach, revealed a clear biphasic profile of lipids. However, these analyses were carried out using a commercial

ELISA kit, which contrasts with data in the paper, which was by LC-MS/MS. We are including these data here for the benefit of the reviewers.

In terms of protein expression requested here and in point 4 below, **Supplementary Figure 5** has been included showing expression of COX-2, mPGES-1 at protein level in immune cells at onset and post-resolution phases as well as EP4 post-resolution macrophages and CD3-positive T.

4. There is similar concern for the expression of the relevant EP receptors, such as EP4, on the surface of the different subsets of macrophages. Once again, the authors over rely on RNA assays as a surrogate for direct expression of EP4 or other EP receptors on the cell surface. Further, the data presented shows increases in EP4 mRNA but these changes are only significant for interstitial macrophages. The only way this data has any import is if the EP4 protein expression is also increased. Further, the authors would need to confirm the converse, activation of this select receptor can promote their selected processed.

Author response: See response to 3 above.

5. While the expression of CCL2 may have been increased in the macrophage populations, epithelial cells are a likely contributor to CCL2. However, all of the assays focus on these cells exclusively. It is likely that local cellular changes in CCL2 expression only modestly impact more global changes in CCL2 in the lung compartment. Total CCL2 assays should be performed.

Author response: We have carried out numerous studies to locate the source of CCL2, especially at day 14 post *S. pneumoniae* injection. While we started with imaging techniques, exhaustive characterisation efforts left us with the conclusion that despite using multiple commercial antibodies and appropriate controls, there was simply too much non-specific binding leaving us unconvinced and unhappy with the data. This was also the case with spectral flow and western blotting techniques. This seem to be something specific to day 14 post-resolution samples where both non-specific binding and auto-fluorescence caused significant and unforeseen challenges. It's for this reason that we spent a substantial amount of time being 100% confident with the imaging data that we have provided. We reached a point where we had to abandon efforts to ensure this paper achieved timely publication.

Minor comments

1. More specific details are required for the *S. pneumoniae* dosing studies. For example, was the Strep administered from frozen aliquots at the established dose or was the strep actively in log phase growth when administered to the mice. This is a subtle detail but maybe critical in observations.

Author response: Bacterial stocks of *S. pneumoniae* were grown and supplied by Jeremy Brown (UCL Respiratory). Briefly, bacterial colonies were isolated from individual CFUs grown on blood agar (Tryptic Soy Agar (Becton Dickinson) supplemented with 3% volume/volume (v/v) defibrinated horse blood) plates and incubated in Tryptic Soy Broth (TSB; Becton Dickson) at 37°C with caps unscrewed (if incubator is 5% CO₂) until the optical density (OD) was 0.3-0.5. Concentration of CFU was determined using OD values and a standard curve. Bacteria were supplemented with 15% glycerol and stored at -80°C until use. This has been included on page 14 of the revised manuscript.

2. In Figure 8B, Ashcroft scores were provided in the figures; however, there is no apparent quantitation of these scores for differences in the sample groups. Further I do not believe that

the vehicle control adequately represents an Ashcroft Score of 0. An uninjured control should also be provided to show a true “0”.

Author response: Quantification of fibrosis has been included on page 12 of the revised manuscript in reference to **Figure 8B**. In this same amended section, we have referred the reader to **Supplementary Figure 7** (referred to as “D0”) for uninjured, baseline lung data as suggested by this reviewer. We also included reference to the Ashcroft scoring system used in the paper in Materials and Methods, page 17 under title “Immunohistochemistry of lung sections”.

3. It is not clear what the data in Supplementary Figure 4a is providing. Are these quantified values of the prostaglandins and family members via ELISA (Concentrations), qPCR (relative quantities) etc?

Author response: In addition to profiles and spectra of PGE₂ and PGI₂ (6 keto PGF1 α) provided in **Figure 5**, data in **supplementary Figure 4** is the remainder of the comprehensive LC-MS/MS lipidomic data analysis on whole mouse lung included for transparency purposes and for those interested in acute inflammation, resolution and post resolution lipidome, see legend to **Supplementary Figure 4** in the main manuscript, page 32.

4. This manuscript would be greatly improved if histological sections were provided for the MC-21 treatment studies as well to show a physiological effect of monocyte/macrophage depletion.

Author response: These data have been included in **Supplementary Figure 8** and reference made to data on page 12 of the revised manuscript including quantification.

5. MF-498 does not antagonize PGE₂. It is a specific EP4 blocker. Studies should be performed using EP2 and EP4 selective agonist and antagonist in splenocytes to confirm effect.

and

6. While both EP2 and EP4 activate cAMP, EP4 can differentially activate cAMP and PI3K/Akt signalling. Further, it has been reported that activation of the alternative EP4 signalling pathways could outpace the effects of the cAMP induction. Studies should be done to elucidate which aspect of EP4 mediates these effects. These studies could be limited to splenocytes.

Author response to 5 and 6: We fully agree that MF-498 is a specific EP4 antagonist as before the studies in this paper were conducted, we tested its efficacy on EP4 versus EP2 receptors. Data are shown here using mouse splenocytes with equivalent data obtained using mouse peritoneal macrophages. In addition, we found little evidence that the selective EP4 agonist CAY10684 activates PI3K/AKT in this system. However, curious as to whether post-resolution T cell populations may differentially signal *via* cAMP and PI3K/AKT when their EP4 is activated, we found

evidence that some CD4 populations of T cells elaborate PI3K/AKT over cAMP. These data have opened possibilities that post-resolution PGE₂ may drive tissue T cell differentiation

through different signalling pathways. However, these pilot experiments were challenging due to the paucity of T cells populations and the signal required for a reliable readout. Indeed, this requires a dedicated body of research to tease apart conclusively, which would be outside the scope to the current report.

REVIEWER COMMENTS

Reviewer #1 (Remarks to the Author):

Title: "Post-resolution macrophage-derived lipids shapes long-term tissue immunity and integrity"

The revised manuscript is much improved. The authors addressed all questions from the reviewer and provided additional data that strengthens the paper.

Specifically:

Figure 4 Treatment with anti-MC-21 Ab reduced macrophages in the lungs... The authors provided new Fig 6 showing 50% reduction in PGE2 in lung macrophages in mice that received treatment with anti-MC-21 Ab

Figure 5 The mechanism of PGE2 induction in lung macrophages... The author explained the approach that is taken to address this question and expressed commitment to address the mechanism of PGE2 induction in lung macrophages in the future studies. Acceptable

Figure 5 The presence of COX-2 and mPGES1 in alveolar and interstitial macrophages early after infection... The author provided a new Supplementary Figure 5 where expression of COX-2 and of mPGES1 are shown in the lung tissues of infected mice at 24 h post infection and at Day 14. Also, the presence of EP4 protein on macrophages and on T cells is shown in the lungs post infection and during resolution phase. These data confirm the statement regarding the role of prostaglandin biosynthesis after inflammation resolution.

The discussion was shortened, is more focused, and is more suitable for the content of the study.

Page 9, reference to Figure 3D in sentence "Mononuclear phagocytes sorted by FACS from naïve and post-resolution lungs" was replaced with reference to Figure 5D.

Reviewer #2 (Remarks to the Author):

The authors answered appropriately my comments.

Reviewer #3 (Remarks to the Author):

The authors have been very responsive to my comments.

This reviewer still recommends being more specific when discussing PGE2/EP4...These references should be limited to EP4 blockade using a small molecule inhibitor as no direct studies were performed instead of stating "PGE2/EP4 blockade".

I am puzzled by the response to #6. The data shows that EP4 antagonism with MF-498 had no effect on EP4-specific agonists (CAY10684). Further, the EP4 agonists had no effect on cAMP induction. None of this data makes sense and does not convince this reviewer that the mechanism they are proposing is active.

As the predominant mechanism of action of the EP2/4 receptor, relayed to the reviewer, is cAMP; how do the authors reconcile their response to Reviewer 3, Comment #6?

This data would be more compelling if they showed the MF-498 reduced cAMP in PGE2-treated samples.

REVIEWERS 1 & 2

We are very pleased to have satisfactorily addressed the issues raised by these reviewers.

REVIEWER 3

1. Being more specific when discussion PGE₂/EP4.

Author response: We have amended the manuscript delineating between PGE₂ synthesis and EP4 antagonism, see track changes.

2. Confusion regarding point 6 under minor comments

Author response: The confusion expressed by this reviewer regarding the effects of EP2 and EP4 agonists with/without MF-498 (EP4 antagonist) is surprising to us. It's possible that this misunderstanding arose from how we labelled the X axis in Figure 1 below, which was in the latest rebuttal letter (submitted October 16th). Specifically, it may be unclear which samples were pre-incubated with MF-498 to antagonise EP4. For this reason, we have altered the labelling to make it clearer, please see Figure 2. Here, there are six treatments. The first and second are cells alone followed by cells treated with a stable PGE₂ analogue, respectively. The latter expectedly elevates cAMP (the chosen readout for EP2/4 signalling). Columns three and four show the effect of selective EP2 and EP4 agonists, respectively, on cAMP, which again is elevated.

Column five is cells pre-incubated with MF-498 (EP4 antagonist) and then exposed to 19(R)-hydroxy PGE₂ (EP2 agonist). Here, the elevation in cAMP caused by 19(R)-hydroxy PGE₂ was not affected by MF-498; this is to be expected as MF-498 is a selective EP4 antagonist.

However, the final column shows cells pre-incubated with MF-498 and then exposed to CAY10684 (EP4 agonist). Here, the elevation in cAMP caused by CAY10684 was expectedly reversed by MF-498 as it's a selective EP4 antagonist.

We hope this is now clearer.

REVIEWER COMMENTS

Reviewer #3 (Remarks to the Author):

I thank the reviewers for providing more clarity to the provided figure.

However, the question remains. If the mechanism of action for PGE2 via EP2 or EP4 is via induction of cAMP, how do the authors reconcile that EP4 blockade does not affect PGE2-mediated cAMP induction?

What is the proposed mechanism of action? In the absence of a clearer explanation, this effect appears to be an artifact of a small molecule inhibitor and may be independent of EP4.

It was asked prior if this could be mediated via blockade of PI3K/Akt induction, which is the unique aspect of EP4. However, the authors stated this effect was modest at best.

Is there a specific cell subtype that is not described that could be providing this unique effect? Some explanation must be provided.

We would like to contextualise the outstanding issues raised to avoid growing confusion and thank reviewer three for their continued patience. The response below pertains to this reviewer only, as reviewers 1 and 2 have no further questions.

As a result of the first round of reviews, we were originally asked to preform “*studies using EP2 and EP4 selective agonist and antagonist in splenocytes to confirm effect*”. Indeed, these experiments have been done, see figure further clarified below and presented in our first and second rebuttal letter. As a reminder, pre-incubating cells with MF-498, the specific EP4 antagonist used in our manuscript, followed by EP2 or an EP4 selective agonists, revealed, as is widely known throughout the scientific literature, that MF-498 blocked the effects of the EP4 agonist (CAY10684) only. We have also added data where we pre-incubated cells with MF-498 before the stable PGE₂ analogue, 11-deoxy PGE₂. From these data MF-498 had negligible effects as PGE₂ will still signal through existing EP2. This is relevant as we discuss below how only EP4 is expressed during post-resolution.

In the second round, we agreed with this reviewer’s comments and were more “*specific when discussing PGE2/EP4*” and altered the discussion section accordingly differentiating between “*PGE₂ synthesis or EP4 antagonism*”.

Also, this reviewer was “*puzzled*” that data in this graph (in rebuttal letters 1 and 2 and below) “*shows that EP4 antagonism with MF-498 had no effect on EP4-specific agonists (CAY10684)*”, reiterating in the same paragraph that MF-498 has “*no effect on cAMP induction*”. In very clear defence of our data, as underlined in response one above, it is obvious that the elevation in cAMP caused by CAY10684 (EP4 receptor agonist) is reversed by MF-498 (as it’s a selective EP4 antagonist), $p < 0.0001$.

Having hoped that this was sufficient to allay any further concerns, this reviewer again questions how we “*reconcile that EP4 blockade does not affect PGE2-mediated cAMP induction?*” We have explicitly and clearly demonstrated this in data presented in rebuttal letters 1 and 2 and in figure below.

It is then stated that “*data would be more compelling if they showed the MF-498 reduced cAMP in PGE2-treated samples*”. As stated above, we have added data where we pre-incubated cells with MF-498 before the stable PGE₂ analogue, 11-deoxy PGE₂. From these data, and contrary to the reviewer’s expectation, MF-498 had little effect on PGE₂-mediate cAMP induction. We interpret these data as the stable PGE₂ analogue being still able to signal through existing EP2. This is relevant as we discuss below how only EP4 is expressed during post-resolution.

In the third round, reviewer three asks “*If the mechanism of action for PGE2 via EP2 or EP4 is via induction of cAMP, how do the authors reconcile that EP4 blockade does not affect PGE2-mediated cAMP induction?*” This is very confusing as we have not included data looking at the effect of EP antagonists on PGE₂ using cAMP as a readout at that point. That said, and as detailed above, we have now added data where we pre-incubated cells with MF-498 before the stable PGE₂ analogue, 11-deoxy PGE₂ with MF-498 having negligible effects as PGE₂ will still signal through EP2. Again, as EP2 is not, or very lowly expressed, during post-resolution biology, the modulatory role of PGE₂ is exerted through EP4, as EP4 is the only PGE₂ receptor expressed during this phase.

Splenocytes and naïve peritoneal macrophages, the model system used in the data shown below, have roughly equivalent levels of EP2 and EP4. Post-resolution biology seems to dampen EP2 receptor expression.

They also question the mode of action of post-resolution PGE₂, specifically “*What is the proposed mechanism of action? In the absence of a clearer explanation, this effect appears*

to be an artifact of a small molecule inhibitor and may be independent of EP4". They also asked whether "this could be mediated via blockade of PI3K/Akt induction, which is the unique aspect of EP4".

From the data in the manuscript and in the figure below, it's unlikely that our data might be an artifact independent of EP4. We show that MF-498 is a clear and obvious antagonist of EP4 **only**, despite the assertions to the contrary by this reviewer of this fact, shown here and extensively published elsewhere. As stated, it must be reminded that EP4 is by far the most predominantly expressed receptor in post-resolution macrophages and T cells, see Figure 5F-G and supplementary data of the manuscript data, with EP2 expression being negligible. In support of this, and in Fig 6-7, the effects of the pan COX inhibitor naproxen are equivalent to that of MF-498. In essence, the post resolution effects of PGE₂ are *via* EP4 only.

In terms of signalling via PI3K/Akt. In this regard, we reiterate our original response to this – "We did find evidence that some CD4 populations of T cells elaborate PI3K/AKT over cAMP. These data have opened possibilities that post-resolution PGE₂ may drive tissue T cell differentiation through different signalling pathways. However, these pilot experiments were challenging due to the paucity of T cells populations and the signal required for a reliable readout. Indeed, this requires a dedicated body of research to tease apart conclusively, which would be outside the scope to the current report."

Therefore, to alleviate what we suspect maybe the underlying concern, we suggest including this sentence in the discussion section, page 25 highlighted in red:

EP4 is the most predominant prostaglandin receptor expressed during post-resolution modulating T cells and preventing tissue damage, with EP1-3 being negligible. The functional role of EP4 is evidenced following the actions of the EP4 antagonist, MF-498 and the pan cyclooxygenase inhibitor naproxen. Collectively, these data suggest that PGE₂ exerts its post-resolution effects through EP4. In terms of downstream signalling, while cAMP is classically activated by EP4 (insert reference here), there is also evidence of PI3K/AKT involvement in EP4 signalling, (insert reference here). As the post-resolution immune landscape is littered with many subtypes of mononuclear phagocytes and T cells, discerning the precise downstream signalling pathway transduced by EP4 is likely cell type specific and tissue niche specific and therefore beyond the scope of this report.

REVIEWERS' COMMENTS

Reviewer #3 (Remarks to the Author):

I would like to thank the authors for the amount of time and effort that they have invested in this project.

This reviewer's central concern is that they have not clearly shown that EP2 is not contributing to the effect and that the effect they are observing may be off-target, as happens with small molecule inhibitors.

While EP4 expression is shown in strep injury in the supplemental figures, no functional data is provided showing that EP4 is upregulated. This is due in part to the overreliance on RNA data, which can be disconnected from functional protein. Further, if EP2 is functionally expressed in normalcy, as shown in the communication to the reviewers, where is the data showing that its expression or function is affected by strep infection?

I minimally ask that the authors confirm that EP2 protein is low and/or reduced in the context of the 14d post-strep.